# GUIDED FLOW POLICY:
# LEARNING FROM HIGH-VALUE ACTIONS
# IN OFFLINE REINFORCEMENT LEARNING

**Franki Nguimatsia Tiofack**[1,*]    **Théotime Le Hellard**[1,*]    **Fabian Schramm**[1,*]
**Nicolas Perrin-Gilbert**[2]    **Justin Carpentier**[1]

[1]Inria and DI-ENS, PSL Research University    [2]Sorbonne Université, CNRS, ISIR, France
[*]Equal contribution. Correspondence to: `franki.nguimatsia-tiofack@inria.fr`

## ABSTRACT

Offline reinforcement learning often relies on behavior regularization that enforces policies to remain close to the dataset distribution. However, such approaches fail to distinguish between high-value and low-value actions in their regularization components. We introduce Guided Flow Policy (GFP), which couples a multi-step flow-matching policy with a distilled one-step actor. The actor directs the flow policy through weighted behavior cloning to focus on cloning high-value actions from the dataset rather than indiscriminately imitating all state-action pairs. In turn, the flow policy constrains the actor to remain aligned with the dataset's best transitions while maximizing the critic. This mutual guidance enables GFP to achieve state-of-the-art performance across 144 state and pixel-based tasks from the OGBench, Minari, and D4RL benchmarks, with substantial gains on suboptimal datasets and challenging tasks.
Webpage: simple-robotics.github.io/publications/guided-flow-policy

## 1 INTRODUCTION

Offline Reinforcement Learning (RL) aims to learn effective policies from static datasets without further interaction with the environment (S. Lange, 2012; Ernst et al., 2005). This paradigm is essential in domains such as robotics and logistics, where online exploration can be unsafe or costly. However, standard off-policy algorithms such as DDPG (Lillicrap et al., 2015) and SAC (Haarnoja et al., 2018), which are successful in online RL, tend to underperform in offline settings since the RL agent cannot interact with the environment. The main challenge is extrapolation error, corresponding to the inability to properly evaluate out-of-distribution actions (Wu et al., 2019; Fujimoto et al., 2019; Kumar et al., 2019; 2020).

Two main lines of work have been proposed to address this challenge. The first one focuses on learning a critic without querying the values of actions outside the dataset (Kostrikov et al., 2021; Nair et al., 2020). The second one, known as the Behavior-Regularized Actor–Critic (BRAC) family[1], mitigates these errors by forcing the learned policy to stay "close" to the unknown behavior policy that generated the dataset (Fujimoto & Gu, 2021; Tarasov et al., 2023; Jaques et al., 2019; Laroche et al., 2019; Wu et al., 2019). The key idea is that out-of-distribution state–action pairs are especially vulnerable to Q-value overestimation, while staying near the empirical distribution reduces extrapolation errors. Minimalist variants achieve this by simply adding a behavior cloning (BC) loss to the policy and/or value updates with respect to dataset actions (Fujimoto & Gu, 2021; Tarasov et al., 2023). Although this approach improves stability, it also raises a trade-off: regularizing too strictly to a potentially suboptimal dataset action may restrict the policy from exploiting higher-reward actions contained in the dataset.

Until recently, most offline RL algorithms were based on Gaussian policies, with limited modeling capacities. Recent development of flow and diffusion-based expressive models (Lipman et al., 2022; Ho et al., 2020; Song et al., 2020), led to new RL algorithms to capture complex and multimodal

---

[1]The abbreviation BRAC refers to the family of methods, not solely the BRAC paper (Wu et al., 2019).

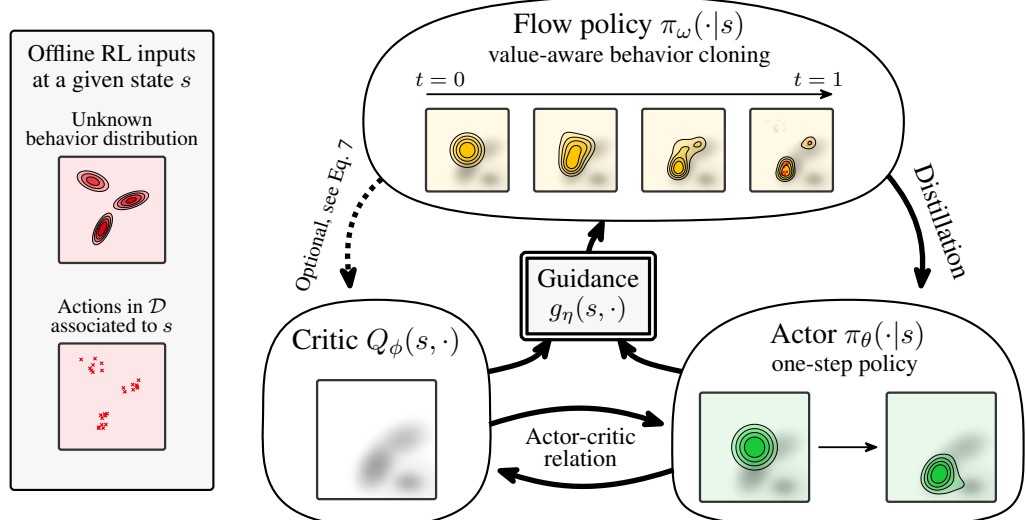

Figure 1: **Overview of the Guided Flow Policy framework.** GFP consists of three main components: (i) in yellow, VaBC, a multi-step flow policy $\pi_\omega$ trained via weighted BC using the guidance term $g_\eta$, (ii) in green, a one-step actor $\pi_\theta$ distilled from the flow policy, and (iii) in gray, a critic $Q_\phi$ guiding action evaluation. $\pi_\omega$ regularizes the actor toward high-value actions from the dataset $\mathcal{D}$; in turn, the actor shapes the flow and optimizes the critic following the actor–critic approach. The different components of the figure are introduced throughout the paper. Each drawing represents the probability distribution of actions $a \in \mathcal{A}$ of a policy, in a current state $s$, except for the gray ones, where it is the value of actions $a \in \mathcal{A}$ in state $s$, according to the critic.

action distributions (Chi et al., 2023; Janner et al., 2022; Wang et al., 2022; Zhang et al., 2025). However, they come at the risk of high computational overhead: iterative sampling slows inference, and directly optimizing the values of output actions would result in unstable backpropagation through time (BPTT). Among recent approaches to address these challenges, Park et al. (2025) proposed a BRAC method with a flow-matching BC model distilled into a one-step policy that also optimizes the critic, enabling expressive policy learning while avoiding BPTT and iterative sampling at inference. Despite these advances, a central limitation remains: the flow-based BC component, similar to standard BC, does not incorporate reward information.

We propose **Guided Flow Policy (GFP)**, a dual-policy BRAC framework with a bidirectional guidance mechanism between a multi-step flow-matching policy, termed Value-aware Behavior Cloning (VaBC), and a distilled one-step actor. VaBC acts as a distributional regularizer for the actor, encouraging it to remain within the support of the behavior policy. VaBC is trained via a weighted-BC mechanism, close to Peng et al. (2019); Zhang et al. (2025), but leveraging the actor and the critic to prioritize cloning high-value actions from the dataset. Unlike previous BRAC approaches, in which the BC regularization indiscriminately treats all state-action pairs (Fujimoto & Gu, 2021; Park et al., 2025), VaBC is a value-aware regularizer integrated into a BRAC approach. In turn, the actor optimizes the critic while being distilled toward VaBC, allowing it to align with the dataset's high-value actions in a given state while maximizing expected returns. Fig. 1 illustrates the GFP framework and Tab. 1 shows how it differs in the regularization mechanisms compared to other BRAC methods.

Our contributions are threefold: (i) we introduce Guided Flow Policy, a simple yet effective BRAC method that integrates value-awareness in the regularization term via a jointly trained weighted flow BC policy, thereby regularizing the actor with the most promising transitions of the dataset; (ii) we extensively evaluate GFP on 144 tasks from standard offline RL benchmarks, showing strong performances with substantial gains on suboptimal datasets and challenging tasks compared to prior works; (iii) we re-assess two previous state-of-the-art offline RL algorithms on these benchmarks, highlighting the critical role of hyperparameter choices and subtle implementation details, aligned in the spirit with the retrospective analysis provided in Tarasov et al. (2023).

## 2 BACKGROUND

**Actor-critic framework in RL.** RL problems are typically formalized as a Markov Decision Process (MDP) (Sutton et al., 1998; Konda & Tsitsiklis, 1999), defined by the tuple $(\mathcal{S}, \mathcal{A}, p, r, \rho, \gamma)$. Here, $\mathcal{S}$ denotes the state space, $\mathcal{A}$ the action space, $p$ the transition dynamics, $r$ the reward function, $\rho$ the initial state distribution, and $\gamma \in [0, 1)$ the discount factor. The behavior of the agent is

Table 1: **Overview of regularization mechanisms within the BRAC framework.**

| | Regularization target | Value-aware regularization | Expressive variant |
|---|---|---|---|
| TD3+BC (Fujimoto & Gu, 2021) | | | |
| ReBRAC (Tarasov et al., 2023) | ↪ dataset actions | ✗ | Diffusion-QL (Wang et al., 2022) |
| FQL (Park et al., 2025) | ↪ learned behavior cloning policy | ✗ | ✓ |
| **GFP (ours)** | ↪ learned value-aware behavior cloning policy | ✓ | ✓ |

governed by a policy $\pi$, mapping states to probability distributions over actions. The objective is to maximize the expected discounted return $\mathbb{E}_{a_t \sim \pi(\cdot|s_t)} \left[ \sum_{t=0}^{\infty} \gamma^t r(s_t, a_t) \right]$, i.e., the expected cumulative reward when following $\pi$ in the MDP. In actor-critic approaches, the policy $\pi$, also referred to as the actor, is trained jointly with a critic $Q$, which approximates the state–action value function. The $Q$-function is defined as $Q^\pi(s, a) = \mathbb{E}_{a \sim \pi(\cdot|s)} \left[ \sum_{t=0}^{\infty} \gamma^t r(s_t, a_t) \mid s_0 = s, a_0 = a \right]$, estimating the expected return after taking action $a$ in state $s$ and subsequently following $\pi$.

Both actor and critic are parametrized as neural networks, with parameters $\theta$ and $\phi$ respectively, and optimized by alternating gradient descent steps on the two objectives:

$$\mathcal{L}^{\mathcal{A}}(\theta) = \mathbb{E}_{s \sim \mathcal{D}, a_\theta \sim \pi_\theta(\cdot|s)} \left[ -Q_\phi(s, a_\theta) \right], \tag{1}$$

$$\mathcal{L}^{\mathcal{C}}(\phi) = \mathbb{E}_{(s,a,r,s') \sim \mathcal{D}, a' \sim \pi_\theta(\cdot|s')} \left[ \left( Q_\phi(s, a) - r - \gamma Q_{\bar{\phi}}(s', a') \right)^2 \right], \tag{2}$$

where $\mathcal{L}^{\mathcal{A}}$ and $\mathcal{L}^{\mathcal{C}}$ refer to the actor and critic losses, respectively, and $\mathcal{D}$ is the set of transitions $(s, a, r, s')$ collected during training. $Q_{\bar{\phi}}$ denotes a second target $Q$-function parameterized by a slowly updated set of weights $\bar{\phi}$, maintained via Polyak averaging, a common stabilization technique in actor–critic methods.

**Minimalist approaches in offline RL.** In offline RL, the agent learns exclusively from a static dataset $\mathcal{D}$, consisting of transitions $(s, a, r, s')$ generated by an unknown behavior policy. In a given state $s$, the distribution of actions of such a behavior policy is illustrated on the left of Fig. 1. This introduces a key challenge compared to the online setting: the distributional shift (S. Lange, 2012; Kumar et al., 2019; Konda & Tsitsiklis, 1999). Indeed, since the learned policy $\pi_\theta$ may select actions outside the dataset's support, value estimates for such out-of-distribution actions can be inaccurate (Kumar et al., 2020; Fujimoto & Gu, 2021). The BRAC approach addresses this issue by constraining the policy $\pi_\theta$ to remain close to the behavior policy (Jaques et al., 2019; Kumar et al., 2019). As emphasized in Fujimoto & Gu (2021), a simple and effective choice is to add a BC term directly into the actor objective. Incorporating this into the actor–critic framework, the actor loss in Eq. 1 becomes:

$$\mathcal{L}^{\mathcal{A}}(\theta) = \mathbb{E}_{(s,a) \sim \mathcal{D}, a_\theta \sim \pi_\theta(\cdot|s)} \left[ -Q_\phi(s, a_\theta) + \alpha \overbrace{\|a_\theta - a\|^2}^{\text{BC term}} \right], \tag{3}$$

where $\alpha$ is a hyperparameter that balances between exploiting high $Q$-values and staying close to the behavior policy. This objective encourages actions that both achieve high-expected returns and remain within the support of the dataset. The critic loss in Eq. 2 remains unchanged.

**Behavior cloning with flow matching.** Flow Matching (FM) (Lipman et al., 2022) is a generative modeling framework that learns a continuous-time transformation, or flow, which maps a simple base distribution (in this work, a standard Gaussian) to a target data distribution. This transformation is defined through a family of intermediate, time-dependent distributions governed by an ordinary differential equation (ODE).

In the context of BC, FM is extended to a conditional setting, where the goal is to approximate a behavior policy $\pi_\omega$ underlying the dataset $\mathcal{D}$. This is achieved by learning a state and time dependent velocity field $v_\omega : [0, 1] \times \mathcal{S} \times \mathbb{R}^d \to \mathbb{R}^d$ that governs the dynamics of a flow, where $d$ is the action dimension. This flow $\psi_\omega(t, s, z)$ is the solution of the family of ODEs characterized by:

$$\forall s \in \mathcal{S}, \quad \frac{d}{dt} \psi_\omega(t, s, z) = v_\omega(t, s, \psi_\omega(t, s, z)), \quad \psi_\omega(0, s, z) = z. \tag{4}$$

This flow, conditioned on the state $s$, maps noise samples $z \sim \mathcal{N}(0, I_d)$ into actions distributed according to $\pi_\omega(\cdot \mid s)$. While sophisticated conditioning strategies can help enhance expressiveness

---

**Algorithm 1: Guided Flow Policy (GFP)**

1  **function** *Integrate* $\mu_\omega(s, z)$
     // Explicit discrete Euler integration with $M$ steps
2     **for** $t = 0, 1, \ldots, M-1$ **do**
3         $z \leftarrow z + \frac{1}{M} v_\omega(t/M, s, z)$
4     **return** $z$

5  **while** *not converged* **do**
6     Sample $\{(s, a, r, s')\} \sim \mathcal{D}$

     // Step 1 -- Train critic $Q_\phi$
7     $z' \sim \mathcal{N}(0, I_d), \quad a' = \mu_{\bar\theta}(s', z')$
8     Update $\phi$ to minimize $\mathbb{E}[(Q_\phi(s,a) - r - \gamma Q_{\bar\phi}(s', a'))^2]$

     // Step 2 -- Train the distilled one-step actor $\pi_\theta$
9     $z \sim \mathcal{N}(0, I_d), \quad a^{\pi_\theta} = \mu_\theta(s, z),$
10    $a^{\pi_{\bar\omega}} = \mu_{\bar\omega}(s, z)$        // Using the Integrate-$\mu_\omega$ function, Line. 1
11    Compute $\lambda = \frac{1}{\frac{1}{N}\sum |Q_\phi(s, a^{\pi_\theta})|}$        // Stop gradient $a^{\pi_\theta}$
12    Update $\theta$ to minimize $\mathbb{E}[-\lambda Q_\phi(s, a^{\pi_\theta}) + \alpha \|a^{\pi_\theta} - a^{\pi_{\bar\omega}}\|_2^2]$

     // Step 3 -- Train the value-aware BC policy $\pi_\omega$
13    Compute $g_\eta(s, a) = \frac{\exp\left(\frac{\lambda}{\eta} Q_\phi(s, a)\right)}{\exp\left(\frac{\lambda}{\eta} Q_\phi(s, a^{\pi_\theta})\right) + \exp\left(\frac{\lambda}{\eta} Q_\phi(s, a)\right)}$    // Stop gradient $a^{\pi_\theta}$
14    $a_t = (1-t)\epsilon + ta$, with $\epsilon \sim \mathcal{N}(0, I_d)$ and $t \sim \mathcal{U}([0,1))$
15    Update $\omega$ to minimize $\mathbb{E}\left[g_\eta(s, a)\|v_\omega(t, s, a_t) - (a - \epsilon)\|_2^2\right]$

**Output:** $\pi_\theta, \pi_\omega, Q_\phi$

---

(e.g., classifier-free guidance (Ho & Salimans, 2022)), we adopt in this work the simplest variant of conditional flow matching (Holderrieth & Erives, 2025). We further employ the optimal transport variant of FM, which uses linear interpolation with uniformly sampled time points (Lipman et al., 2022). For $(s, a) \sim \mathcal{D}$, $\epsilon \sim \mathcal{N}(0, I_d)$, and $t \sim \mathcal{U}([0, 1])$, we define the interpolated point $a_t = (1 - t)\epsilon + ta$, whose target velocity is $a - \epsilon$. The velocity field $v_\theta$ is then trained by least-squares regression toward this reference, yielding the conditional flow-matching BC loss (Holderrieth & Erives, 2025):

$$\mathcal{L}^{\text{FM-BC}}(\omega) = \mathbb{E}_{(s,a)\sim\mathcal{D}, \epsilon\sim\mathcal{N}(0, I_d), t\sim\mathcal{U}([0,1])}\left[\|v_\omega(t, s, a_t) - (a - \epsilon)\|_2^2\right]. \tag{5}$$

Once the velocity field is learned, the corresponding flow $\psi_\omega : [0, 1] \times \mathcal{S} \times \mathbb{R}^d \to \mathcal{A}$ defines an approximation of the behavior policy. At inference, an action is obtained by sampling a random noise $z \sim \mathcal{N}(0, I_d)$ and integrating the flow from 0 to 1 using an ODE solver (e.g., an explicit Euler method). We denote by $\mu_\omega(s, z) := \psi_\omega(1, s, z)$ the value of the integrated flow at time 1. In this way, behavior cloning can be naturally expressed as conditional flow matching in the action space.

**Flow policy for offline RL.** Following the idea of Diffusion Q-Learning by Wang et al. (2022), a straightforward way to train a flow policy for offline RL is to replace the BC term in the actor loss (Eq. 3) with the flow-matching BC loss (Eq. 5). However, the iterative sampling procedure makes training expensive, due to recursive backpropagation through time (BPTT) in the actor loss, and also results in a slow inference at test time. To mitigate these limitations, Park et al. (2025) suggests distilling the iterative flow-matching BC policy into a one-step policy that also maximizes the critic.

## 3 GUIDED FLOW POLICY

We now detail the GFP algorithm that builds on top of Fujimoto & Gu (2021); Park et al. (2025). GFP integrates a Value-aware Behavior Cloning (VaBC) flow policy with a distilled one-step actor through bidirectional guidance. VaBC leverages the actor and the critic to selectively clone high-value dataset actions, providing more targeted regularization than in standard BRAC approaches. The distilled actor, in turn, maximizes the critic while avoiding BPTT and iterative sampling. GFP is composed of three main components: the critic $Q_\phi$, the actor $\pi_\theta$, and the VaBC policy $\pi_\omega$. The complete algorithm is presented in Algo. 1, and the approach is illustrated in Fig. 1.

**Step 1 – Learning the critic $Q_\phi$.** The critic is trained using the Bellman mean-squared loss:

$$\mathcal{L}^{\mathcal{C}}(\phi) = \mathbb{E}_{(s,a,r,s')\sim\mathcal{D},\, a'\sim\pi_\theta(\cdot|s')}\left[\left(Q_\phi(s, a) - \underbrace{(r + \gamma Q_{\bar\phi}(s', a'))}_{\text{Bellman target } y}\right)^2\right], \tag{6}$$

where $Q_{\bar\phi}$ denotes the target network. $y(s, r, s') := r + \gamma Q_{\bar\phi}(s', a')$ corresponds to the standard Bellman target in actor-critic methods, which we use by default in this work. Yet, since VaBC is designed to prioritize cloning the most promising dataset actions for a given state, we have also

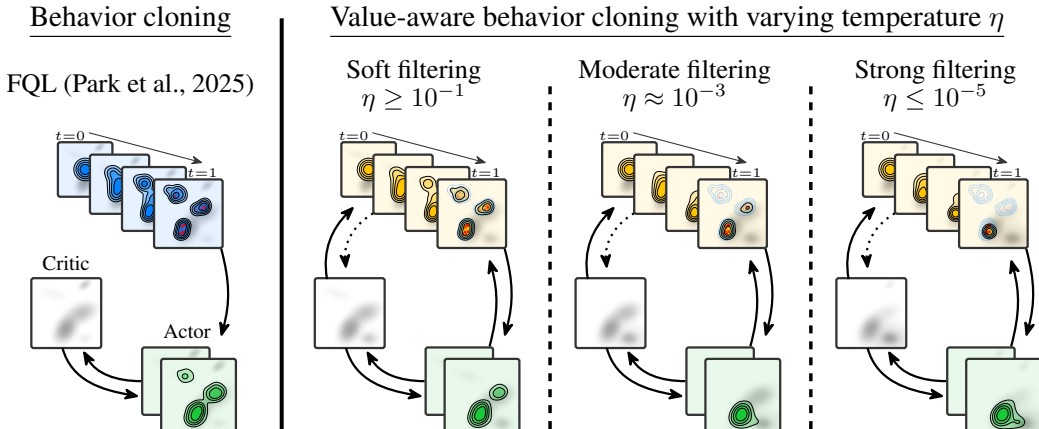

Figure 2: **Comparison of behavior cloning under different levels of guidance. Left:** Prior work (e.g., FQL by Park et al. (2025)) uses no filtering, indiscriminately imitating all state-action pairs. **Right:** In contrast, our method introduces a temperature-controlled guidance mechanism, as shown in Eq. 10, resulting in VaBC. At high temperatures, the guidance is weak, so the actor is influenced by many candidate actions. At moderate temperatures, the filtering becomes sharper, giving more weight to higher-value actions while still keeping enough regularization and exploration. At low temperatures, the filtering is very selective, concentrating almost exclusively on the highest-value actions according to the critic. However, excessive concentration at very low temperatures may allow the actor to escape the dataset's action distribution, as shown on the right in green, leading to critic overestimation and out-of-distribution issues. Importantly, VaBC cannot escape the dataset's action distribution even at very low temperatures, since it trains exclusively on in-distribution state-action pairs. The dashed blue contours in the final yellow drawings (first row) illustrate this constraint.

considered a more conservative variant of the Bellman target:

$$y^{\text{VaBC}}(s, r, s') = r + \tfrac{\gamma}{2}\Big(Q_{\bar{\phi}}(s', \mu_\theta(s', z)) + Q_{\bar{\phi}}(s', \mu_\omega(s', z))\Big), \quad z \sim \mathcal{N}(0, I_d), \qquad (7)$$

where $\mu_\theta(s', z)$ denotes the action from the actor and $\mu_\omega(s', z)$ the action from the VaBC policy. Here, as mentioned in Sec. 2 and outlined in Line 1 of Alg. 1, $\mu_\theta(s, z)$ and $\mu_\omega(s, z)$ are actions sampled from $\pi_\theta(\cdot|s)$ and $\pi_\omega(\cdot|s)$, respectively, with initial input noise $z$. The Bellman target $y^{\text{VaBC}}(s, r, s')$ corresponds to an averaging between two estimates of the Q-value: $Q_{\bar{\phi}}(s', \mu_\theta(s', z))$ which can overestimate the real Q-value; and $Q_{\bar{\phi}}(s', \mu_\omega(s', z))$ which can underestimate the real Q-value. This choice can lead to substantial performance improvements in certain situations, as studied in the appendix, Sec. B.1.

**Step 2 – Learning the actor $\pi_\theta$.** The actor $\pi_\theta$ is trained by behavior regularized policy gradients, to maximize the Q-function while distilling the distribution of valuable actions learned by $\pi_\omega$. This is achieved by minimizing the following objective:

$$\mathcal{L}^{\mathcal{A}}(\theta) = \mathbb{E}_{s \sim \mathcal{D}, z \sim \mathcal{N}(0, I_d)}\Big[ -\lambda Q_\phi\big(s, \mu_\theta(s, z)\big) + \alpha\|\mu_\theta(s, z) - \mu_\omega(s, z)\|_2^2\Big]. \qquad (8)$$

The normalization term $\lambda = \frac{1}{\frac{1}{N}\sum |Q_\phi(s,a)|}$ is based on the average absolute Q-value, estimated over mini-batches rather than over the entire dataset (Fujimoto & Gu, 2021).

The distillation term encourages the actor to stay close to VaBC. In this way, the actor learns to select actions that maximize return while avoiding out-of-distribution actions, as it is constrained to remain near the support of high-value dataset behaviors.

**Step 3 – Learning the flow policy $\pi_\omega$.** The VaBC policy $\pi_\omega$ is optimized via a weighted flow-matching behavior cloning:

$$\mathcal{L}^{\text{VaBC}}(\omega) = \mathbb{E}_{(s,a) \sim \mathcal{D}, \epsilon \sim \mathcal{N}(0, I_d), t \sim \mathcal{U}([0,1])}\Big[ g_\eta(s, a) \|v_\omega(t, s, a_t) - (a - \epsilon)\|_2^2\Big], \qquad (9)$$

where

$$g_\eta(s, a) := \frac{\exp\big(\tfrac{\lambda}{\eta} Q_\phi(s, a)\big)}{\exp\big(\tfrac{\lambda}{\eta} Q_\phi(s, a)\big) + \exp\big(\tfrac{\lambda}{\eta} Q_\phi(s, \mu_\theta(s, z))\big)}, \quad z \sim \mathcal{N}(0, I_d). \qquad (10)$$

Intuitively, for a given state-action pair $(s, a)$ sampled from the dataset $\mathcal{D}$, $g_\eta(s, a)$ compares the quality between the dataset action $a$ and a proposal of the actor $\mu_\theta(s, z)$ in a soft-max approach. If the dataset action has a higher Q-value, this implies that $g_\eta(s, a) > 0.5$, placing greater emphasis on cloning it. Conversely, if the dataset action is worse, $g_\eta(s, a) < 0.5$, then it reduces its influence. This ensures that VaBC selectively clones high-value dataset behaviors. This makes sense because the actor itself is constrained to remain close to the dataset's action distribution. Importantly, VaBC is learned jointly with the actor and the critic, not beforehand.

Here, $\lambda$ is the same Q-normalization factor used in the actor loss, ensuring consistent scaling across components. The parameter $\eta > 0$ is a temperature hyperparameter that controls the sharpness of the weighting: small $\eta$ makes $g_\eta(s, a)$ more selective, while large $\eta$ smooths the weighting. Importantly, since $g_\eta(s, a) \in (0, 1)$, VaBC avoids degeneracy during early training when the critic is unreliable, ensuring stable learning.

The key contribution of GFP is to add value-awareness in the behavior regularization component of a BRAC framework, effectively combining the two predominant policy extraction methods studied by Park et al. (2024b): weighted-BC and behavior regularized policy gradients, further presented in Sec. 5. One could employ an advantage weighted term, similar to AWR (Peng et al., 2019), using the actor to compute a baseline:

$$g_\eta^{\text{AWR}}(s, a) := \exp\left(\tfrac{\lambda}{\eta}\big(Q_\phi(s, a) - Q_\phi(s, \mu_\theta(s, z))\big)\right) \tag{11}$$

As recommended by Peng et al. (2019), in that case, a clipping term should be added for stability. Instead, we propose the guiding function $g_\eta$, which corresponds to a soft-max between $Q_\phi(s, a)$ and $Q_\phi(s, \mu_\theta(s, z))$. In appendix, Sec. B.3, Tab. 7 reports results when using $g_\eta^{\text{AWR}}$ instead of $g_\eta$, over 65 tasks.

**Analysis of the temperature in the guidance function** $g_\eta$**.** In Fig. 2, we illustrate how the temperature parameter controls value-guided filtering, balancing dataset fidelity with value exploitation. Lower temperatures sharpen the filter, shifting the policy from broadly imitating the dataset to emphasizing higher-value actions. Moderate values achieve the best trade-off, prioritizing promising actions while preserving diversity. In contrast, excessively low temperatures over-concentrate the VaBC policy, destabilizing training and degrading the critic by pushing the actor out of distribution.

## 4 EXPERIMENTS

### 4.1 MAIN RESULTS: EXTENSIVE OFFLINE RL BENCHMARKS

**Benchmarks.** We conducted extensive experiments over a suite of robot locomotion and manipulation tasks, spanning three major benchmarks: D4RL (Fu et al., 2020), its successor Minari (Younis et al., 2024), and the recently proposed OGBench (Park et al., 2024a). For comparability with existing works, we first evaluate on D4RL's AntMaze (6 tasks) and Adroit (12 tasks). We also present results on Minari, evaluating both GFP and FQL, to facilitate the community's migration from D4RL. Minari includes all available Gym-Mujoco datasets (Hopper, HalfCheetah, and Walker, each with 3 tasks), as well as Adroit (12 tasks). Our most extensive evaluation focuses on OGBench, which offers substantially more complex and challenging tasks than D4RL. Following Park et al. (2025), we use the reward-based single-task variants of OGBench. We evaluate across 9 locomotion and 11 manipulation environments, each with 5 tasks, for a total of 100 state-based tasks, and we also consider 5 pixel-based tasks. Combined with D4RL and Minari, we benchmark GFP on 144 tasks overall, including evaluation of prior methods, we did about 15 000 runs in total. Our JAX-based implementation of GFP will be released after the rebuttal phase. It can complete one training run in under 30 minutes on modern GPUs.

**Comparison against previous works.** We first compare GFP against 10 prior methods, including standard offline RL, flow, and diffusion-based methods on the 50 OGBench tasks reported by Park et al. (2025) (see Sec. 5 and Fig 5 for a presentation of prior works, Tab. 10 for the full results). Performance profiles synthesizing these comparisons are presented Fig. 3a, following Agarwal et al. (2021). GFP clearly stands out compared to all these prior works. We also compare against 6 methods on D4RL (Tab. 11) and evaluate both GFP and FQL on Minari (see Tab. 12) to facilitate the community's migration from D4RL.

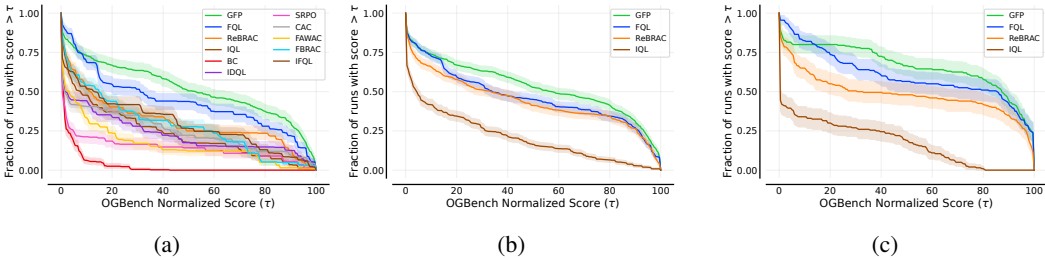

Figure 3: **OGBench analysis.** (a) Performance profiles for 50 tasks comparing GFP against a wide range of prior works, showing the fraction of tasks where each algorithm achieves a score above threshold $\tau$, using the evaluation reported by Park et al. (2025). (b) Performance profiles on 105 tasks, including more challenging ones, and carefully reevaluated prior methods. (c) Performance profiles restricted to 30 noisy and explore tasks.

**Extensive study on 144 tasks.** GFP is further evaluated against 3 baselines: (i) FQL (Park et al., 2025), as it comes second only to GFP on the first 50 tasks; (ii) ReBRAC (Tarasov et al., 2023), as we were able to improve its performance compared to previously reported results substantially, see Sec. 4.3, and (iii) IQL (Kostrikov et al., 2021) to represent in-sampling approaches. Tab. 2 summarizes our results grouped by environment, with detailed per-task results in the appendix (Tabs. 8, 9, 11, and 12). Together with performance profile plots Figs. 3b and 3c, it confirms that GFP achieves state-of-the-art performance, with particularly substantial gains on noisy and challenging environments. For instance, on the cube-double-noisy and cube-triple-noisy datasets, GFP achieves an average score of 63 and 24, respectively, compared to 38 and 4 for FQL, and 20 and 5 for ReBRAC. Similarly, GFP stands out in some very challenging locomotion tasks, such as humanoidmaze-large-navigate (18 vs. 7 for FQL and 13 for ReBRAC), and manipulation tasks, such as cube-triple-play (16 vs. 4 for FQL and 3 for ReBRAC).

Table 2: **Offline RL results.** GFP achieves best or near-best performance on all 144 benchmark tasks. Results are averaged over 8 seeds for state-based tasks and 4 seeds for pixel-based ones, with values reported from prior works (Park et al., 2025; Tarasov et al., 2023; Fu et al., 2020) in *italic*, and values within 95% of the best performance are shown in bold. GFP actor $\pi_\theta$ represents our primary policy, while GFP VaBC $\pi_\omega$ is reported as a byproduct of the training procedure. Full per-task results are provided in the appendix Tabs. 8, 9, 11, and 12, and the comparison with 10 methods on the 50 previously evaluated OGBench tasks Tab. 10.

| Task Category | Offline RL algorithms | | | | |
|---|---|---|---|---|---|
| | IQL | ReBRAC | FQL | **GFP** actor $\pi_\theta$ | **GFP** VaBC $\pi_\omega$ |
| OGBench antmaze-large-navigate-singletask (5 tasks) | *53 ± 3* | **95.9 ± 0.4** | 88.1 ± 3.4 | **93.8 ± 1.5** | 90.0 ± 1.3 |
| OGBench antmaze-large-stitch-singletask (5 tasks) | *30.4 ± 3.2* | **89.2 ± 6.6** | 58.1 ± 8.7 | 68.9 ± 0.8 | 57.6 ± 3.2 |
| OGBench antmaze-large-explore-singletask (5 tasks) | *12.9 ± 1.7* | 82.7 ± 7.6 | **87.9 ± 6.6** | **91.9 ± 0.9** | **89.3 ± 1.1** |
| OGBench antmaze-giant-navigate-singletask (5 tasks) | *4 ± 1* | **33.2 ± 5.7** | 16.3 ± 8.2 | 27.9 ± 8.5 | 0.8 ± 0.2 |
| OGBench humanoidmaze-medium-navigate-singletask (5 tasks) | *33 ± 2* | 59.2 ± 12.1 | *58 ± 5* | **72.0 ± 2.8** | 35.9 ± 2.7 |
| OGBench humanoidmaze-medium-stitch-singletask (5 tasks) | *27.3 ± 2.9* | 61.1 ± 8.2 | **63.2 ± 6.7** | **66.2 ± 5.7** | 39.5 ± 2.1 |
| OGBench humanoidmaze-large-navigate-singletask (5 tasks) | *2 ± 1* | 12.9 ± 4.2 | 6.5 ± 2.7 | **17.8 ± 9.6** | 2.4 ± 1.1 |
| OGBench antsoccer-arena-navigate-singletask (5 tasks) | *8 ± 2* | 55.9 ± 1.5 | *60 ± 4* | **57.9 ± 1.9** | 10.3 ± 0.7 |
| OGBench antsoccer-arena-stitch-singletask (5 tasks) | *2.8 ± 1.0* | 22.0 ± 1.5 | 28.6 ± 2.3 | **30.5 ± 2.2** | 1.4 ± 0.3 |
| OGBench cube-single-play-singletask (5 tasks) | *83 ± 3* | *91 ± 2* | *96 ± 1* | **98.8 ± 0.4** | 39.7 ± 4.1 |
| OGBench cube-single-noisy-singletask (5 tasks) | *53.2 ± 4.1* | **98.4 ± 0.6** | **100.0 ± 0.0** | **100.0 ± 0.0** | **99.9 ± 0.1** |
| OGBench cube-double-play-singletask (5 tasks) | *7 ± 1* | 12.6 ± 1.8 | *29 ± 2* | **47.2 ± 1.6** | 6.4 ± 1.0 |
| OGBench cube-double-noisy-singletask (5 tasks) | *4.5 ± 0.8* | 19.6 ± 2.1 | 38.2 ± 5.3 | **63.1 ± 3.3** | 9.4 ± 0.8 |
| OGBench cube-triple-play-singletask (5 tasks) | *0.1 ± 0.1* | 2.9 ± 1.2 | 3.9 ± 1.5 | **15.9 ± 2.0** | 7.6 ± 1.6 |
| OGBench cube-triple-noisy-singletask (5 tasks) | *4.8 ± 1.2* | 5.2 ± 2.9 | 3.5 ± 1.6 | **24.5 ± 2.8** | 8.6 ± 1.2 |
| OGBench puzzle-3×3-play-singletask (5 tasks) | *9 ± 1* | *21 ± 1* | *30 ± 1* | 23.1 ± 2.2 | 19.2 ± 2.9 |
| OGBench puzzle-4×4-play-singletask (5 tasks) | *7 ± 1* | 17.1 ± 1.3 | *17 ± 2* | **26.1 ± 2.1** | 9.5 ± 1.1 |
| OGBench puzzle-4×4-noisy-singletask (5 tasks) | *0.1 ± 0.0* | 1.1 ± 0.3 | 15.6 ± 1.1 | **18.8 ± 1.7** | **19.3 ± 1.0** |
| OGBench scene-play-singletask (5 tasks) | *28 ± 1* | 41.6 ± 3.6 | *56 ± 2* | 53.5 ± 2.9 | **57.6 ± 1.7** |
| OGBench scene-noisy-singletask (5 tasks) | *16.0 ± 1.2* | 39.9 ± 2.6 | **59.3 ± 1.4** | **57.5 ± 0.9** | **58.5 ± 1.0** |
| OGBench visual manipulation (5 tasks) | *42 ± 4* | *60 ± 2* | *65 ± 2* | 62.8 ± 1.5 | – |
| D4RL antmaze (6 tasks) | *17* | *76.8* | *84 ± 3* | **83.1 ± 2.7** | 70.2 ± 3.0 |
| D4RL Adroit (12 tasks) | *48* | *59* | *52 ± 1* | 52.8 ± 1.4 | 49.6 ± 1.3 |
| Minari Adroit (12 tasks) | – | – | 40.6 ± 0.4 | **48.3 ± 2.3** | 46.1 ± 1.7 |
| Minari hopper (3 tasks) | – | – | 79.6 ± 10.3 | **91.7 ± 4.5** | **91.5 ± 12** |
| Minari halfcheetah (3 tasks) | – | – | 97.8 ± 2.0 | **109.1 ± 2.0** | 103.1 ± 1.8 |
| Minari walker2d (3 tasks) | – | – | **121.7 ± 1.3** | **124.5 ± 0.8** | **122.2 ± 1.1** |
| **Average OGBench (105 tasks)** | 20.4 | 43.9 | 46.7 | **53.2** | 33.1[2] |
| **Average D4RL (18 tasks)** | 54.0 | 64.8 | 62.1 | **63.0** | 56.5 |
| **Average Minari (21 tasks)** | – | – | 65.9 | **74.1** | 71.6 |

[2] average without pixel-based tasks.

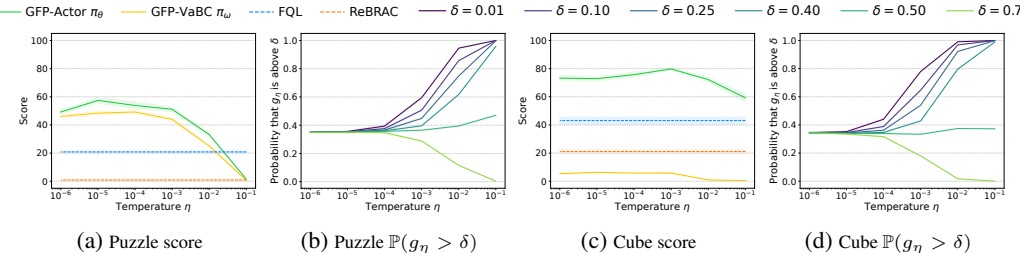

(a) Puzzle score     (b) Puzzle $\mathbb{P}(g_\eta > \delta)$     (c) Cube score     (d) Cube $\mathbb{P}(g_\eta > \delta)$

Figure 4: **Temperature analysis on challenging OGBench Puzzle (left) and Cube (right) tasks with suboptimal data. Plots (a) and (c):** performance scores across temperature values $\eta$ for our GFP method (Actor $\pi_\theta$ and VaBC $\pi_\omega$) compared to baselines (FQL, ReBRAC) on puzzle-4x4-noisy-task3 and cube-double-noisy-task2. **Plots (b) and (d):** probability that the guidance term $g_\eta$ is above different thresholds $\delta$ as a function of temperature, illustrating how temperature controls the sharpness of value-guided filtering.

**Value-aware behavior cloning.** VaBC serves as the regularization component of GFP, encouraging the actor maximizing the value to stay close to some actions from the dataset. While the actor $\pi_\theta$ is GFP primary policy, through our bidirectional training procedure, we obtain the VaBC policy $\pi_\omega$ as a byproduct that can also be exploited and evaluated. As shown in Tab. 2, VaBC achieves good performance across benchmarks while being fundamentally in-sample, justifying its role as a value-aware regularizer for the main actor.

### 4.2 ANALYSIS OF THE TEMPERATURE PARAMETER $\eta$

We investigate the impact of the temperature-controlled guidance mechanism. In Fig. 4, GFP is evaluated with varying temperature $\eta$ on two challenging *noisy* tasks from OGBench, characterized as *highly suboptimal data* according to Park et al. (2024a). The presence of low-quality demonstrations makes selective action emphasis decisive for effective learning. Figs. 4a and 4c demonstrate the advantages of moderate temperatures. Very low temperatures cause training instability due to over-concentration on narrow action sets, while very high temperatures fail to provide sufficient filtering of suboptimal actions. As the temperature decreases, VaBC $\pi_\omega$ performance improves, confirming that the value-guided filtering mechanism successfully emphasizes higher-value actions, until the temperature is too low.

Figs. 4b and 4d illustrate the filtering behavior by showing the probability that the guidance term $g_\eta$ exceeds various thresholds $\delta$ (ranging from 0.01 to 0.75). At extremely low temperatures ($\eta \leq 10^{-5}$), the guidance term exhibits near-binary behavior: any slight differences between $Q(s, a)$, with $a$ from the dataset, and $Q(s, a^{\pi_\theta})$, with $a^{\pi_\theta} \sim \pi_\theta(\cdot|s)$, in state $s$, result in the guidance term approaching either 1 or 0 according to Eq. 10, in this case $\mathbb{P}(g_\eta > 0.75) \approx \mathbb{P}(g_\eta > 0.01)$. As the temperature increases, the filtering becomes softer, creating more gradual transitions in the guidance values. This leads to a broader distribution of filtering probabilities across different thresholds, demonstrating how higher temperatures preserve more of the original dataset diversity. In contrast, lower temperatures create sharper distinctions between high-value and low-value actions. In the appendix, in Sec. B.4, we extend this study to two additional tasks.

**Sensitivity analysis to $\alpha$ and $\eta$.** In Sec. B.5 we evaluate how sensitive GFP is to variations of $\alpha$, the BC coefficient, and $\eta$, on 12 tasks, with comparisons to FQL. It reveals how they relate and demonstrates that GFP is only mostly sensitive to the proper choice of $\alpha$, as for any BRAC method.

### 4.3 RE-EVALUATION OF PRIOR WORKS ON OGBENCH

To obtain a fair comparison of GFP against prior methods, we reevaluate existing baselines on OGBench. During the development of our method, we observed that several task-specific hyperparameters (e.g., discount factor $\gamma$, minibatch size $B$, and critic aggregation scheme for doubled Q-learning (Fujimoto et al., 2018)) have a significant impact on performance. Tab. 3 reports results under these revised settings, showing that careful tuning can substantially improve the reported scores of both ReBRAC and FQL. Since the optimal values for these hyperparameters were generally consistent across methods, we treated them as task-specific and, by default, applied the same settings to GFP, FQL, and ReBRAC (see Tabs. 4 and 5 in the appendix for detailed values).

Table 3: **Impact of task-specific hyperparameters on OGBench performance.**

| Task Environment | Bigger discount factor $\gamma$ | | | |
|---|---|---|---|---|
| | Previously reported in Park et al. (2025) | | Our evaluations | |
| | ReBRAC | FQL | ReBRAC | FQL |
| antmaze-large-navigate (5 tasks) | $81 \pm 5$ | $79 \pm 3$ | $95.9 \pm 0.4$ | $88.1 \pm 3.4$ |
| humanoidmaze-large-navigate (5 tasks) | $2 \pm 1$ | $4 \pm 2$ | $12.9 \pm 4.2$ | $6.5 \pm 2.7$ |
| | **Bigger minibatch size** | | | |
| | Previously reported in Park et al. (2025) | | Our evaluations | |
| | ReBRAC | FQL | ReBRAC | FQL |
| antmaze-giant-navigate (5 tasks) | $26 \pm 8$ | $9 \pm 6$ | $33.2 \pm 5.7$ | $16.3 \pm 8.2$ |
| | **Same critic aggregation for ReBRAC as used in FQL** | | | |
| | Previously reported in Park et al. (2025) | | Our evaluations | |
| | ReBRAC | | ReBRAC | |
| humanoidmaze-medium-navigate (5 tasks) | $22 \pm 8$ | | $59.2 \pm 12.1$ | |
| antsoccer-arena-navigate (5 tasks) | $0 \pm 0$ | | $55.9 \pm 1.5$ | |
| cube-double-play (5 tasks) | $12 \pm 1$ | | $12.6 \pm 1.8$ | |
| scene-play (5 tasks) | $41 \pm 3$ | | $41.6 \pm 3.6$ | |
| puzzle-4x4-play (5 tasks) | $14 \pm 1$ | | $17.1 \pm 1.3$ | |

## 5 RELATED WORK

**Offline RL.** Early methods addressed distributional shift by working on both how the critic is learned and how the policy is extracted. On the critic side, Conservative Q-Learning (CQL) (Kumar et al., 2020) penalizes out-of-distribution value estimates and Implicit Q-Learning (IQL) (Kostrikov et al., 2021) avoids querying such values via expectile regression, see Fig 5a. For policy extraction, weighted behavior cloning treats offline RL as supervised learning, typically using the advantage value as the weight, as in AWR or AWAC (Peng et al., 2019; Nair et al., 2020), see Fig 5d. The policy extraction method found to lead to the best performance and scalability in standard offline-RL is behavior-regularized policy gradient (Fujimoto & Gu, 2021; Park et al., 2024b), abbreviated DDPG+BC and also called reparametized policy gradients, which consists in directly maximizing the value function, while regularizing with a behavior cloning term. The BRAC family studied in this paper uses behavior-regularized policy gradient and trains the critic directly using the actor, see Fig 5c. ReBRAC (Tarasov et al., 2023) demonstrated state-of-the-art performance from this simple idea, through careful hyperparameter tuning and architectural choices.

**Control with diffusion and flow models.** Expressive generative models enable multi-modal action and trajectory distributions modeling and have been integrated into control along several axes. They have been used for trajectory and motion planning, trajectory optimization, hierarchical control (Janner et al., 2022; Ajay et al., 2022; Zheng et al., 2023; Liang et al., 2023; Li et al., 2023; Suh et al., 2023; Venkatraman et al., 2023; Lee et al., 2023; Ma et al., 2024; Chen et al., 2024a; Pan et al., 2024; Le Hellard et al., 2025), as well as for world modeling (Ding et al., 2024b; Alonso et al., 2024; Lee et al., 2024). In offline RL, numerous ideas emerged to integrate diffusion and flow models in the various policy extraction methods.

**Weighted behavior cloning.** A straightforward approach is to add weight in front of the cloning loss of diffusion and flow models, examples include FAWAC (Park et al., 2025), EDP (Kang et al., 2023), and QVPO (Ding et al., 2024a). Related energy-guided methods explicitly target to sample from $\pi_\theta(a \mid s) \propto \mu(a \mid s) \exp(\eta Q(s,a))$, where $\mu$ is the behaviour policy. This includes CEP (Lu et al., 2023) and QIPO (Zhang et al., 2025; Alles et al., 2025), see Fig 5d.

**Behavior-regularized policy gradient.** Combining the most effective policy extraction method with expressive models is a motivating research direction: Diffusion-QL (Wang et al., 2022) (Fig 5c), DiffCPS (He et al., 2023), trust-region formulations for diffusion policies (DTQL) (Chen et al., 2024b), entropy Q-ensemble regularization (Zhang et al., 2024), Consistency-AC with consistency models (Ding & Jin, 2023), and diffusion policy gradients (Li et al., 2024; Yang et al., 2023). In that case, the main challenge is the drawbacks of backpropagation through time (BPTT) when trying to directly optimize the value of the outputs of an iterative process. To avoid BPTT, FQL (Park et al., 2025) (Fig 5e) trains a flow matching with a standard BC loss, and then leverages it in the BC regularization term of a one-step BRAC actor. Beyond these families, other policy extractors include rejection sampling from a fixed BC generative policy: IDQL (Hansen-Estruch et al., 2023) (Fig 5b), DICE (Mao et al., 2024), and SfBC (Chen et al., 2022).

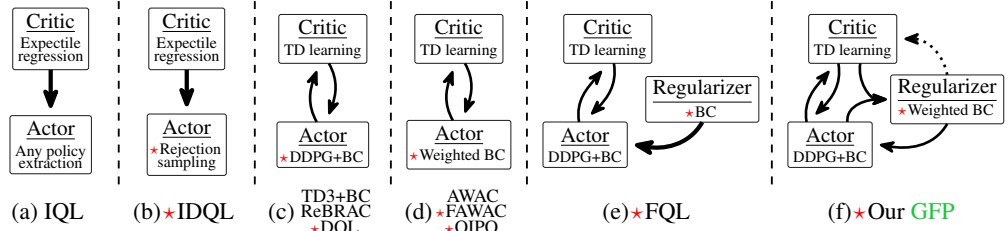

Figure 5: **Overview of some offline-RL frameworks.** The symbol ⋆ indicates the use of a diffusion or a flow model, and specifically in which component. Each box, provides the name of the component (e.g. critic) and the working principle that is used to train it (e.g. TD learning). The arrows indicate how the components depend on each other, while the dashed arrow is optional (see Eq. 7)

**Positioning GFP within prior work.** Our method, GFP (illustrated Fig 1 and Fig 5f), extends the BRAC family of offline RL algorithms and is closely related to FQL (Park et al., 2025), sharing the same goal of avoiding backpropagation through time. However, the distinctive contribution of GFP lies in integrating the weighted BC approach for the regularization component of the BRAC actor. Rather than using weighted BC to train the final actor directly (as in Fig 5d), GFP employs it to inject value-awareness into the regularizer VaBC. Compared to other methods using a weighted-BC policy, VaBC is not GFP's actor, its weights are computed using the one-step actor trained by DDPG+BC, with the latter using VaBC as a regularizer. This bidirectional guidance between the two policies fundamentally distinguishes GFP from other methods that use weighted BC as their main actor, computing the weights based on the weighted BC policy itself, see Fig 5d.

Removing the value-aware guidance from the GFP flow policy yields FQL (Fig 5e), and removing the one-step actor implies using the flow policy to train the critic, yielding FAWAC (Fig 5d). Alternatively, one could use rejection sampling instead of weighted-BC to regularize the actor with the best actions found by a BC policy, yet at the cost of sampling multiple actions from the flow policy for each policy update step.

## 6 DISCUSSION AND CONCLUSION

In this work, we revisited behavior regularization for offline RL. BRAC approaches have been shown to be highly effective for policy extraction (Tarasov et al., 2023; Park et al., 2024b). However, conventional BRAC approaches constrain the learned policy to remain close to the raw dataset distribution. While this reduces instability, the regularization term itself typically does not distinguish between the high and low-value actions. This limitation is problematic in suboptimal datasets, where regularizing with all transitions indiscriminately can hinder performance.

To address this, we introduced Guided Flow Policy (GFP). GFP couples a multi-step flow-matching policy trained with value-aware behavior cloning and a distilled one-step actor through a bidirectional guidance mechanism. GFP leverages the expressiveness of flow policies while adding value awareness directly in the flow part, without the drawbacks of backpropagation through time.

Our analysis provides several insights. First, although standard behavior-regularized actor–critic methods, such as ReBRAC, are competitive with good hyperparameter tuning, their dependence on value-agnostic behavior regularization limits performance on suboptimal datasets. In particular, in prior BRAC approaches, adding low-quality transitions in the dataset will degrade performance, while the value-awareness of VaBC filters them out. Second, while generative models such as flow or diffusion policies can represent dataset distributions more flexibly, they inherit these same limitations if trained to match the raw dataset without explicit guidance. GFP leverages the benefits of two distinct policy extraction methods: the effectiveness and scalability of behavior-regularized policy gradient, and the stability of weighted BC. This combination exploits well the expressivity of flow models, without backpropagating through time. This synergy enables GFP to consistently achieve state-of-the-art results, as demonstrated by our extensive and rigorous study on 144 offline RL tasks.

Nonetheless, GFP depends on the availability of a sufficiently accurate critic. In datasets lacking high-value actions or when the critic cannot reliably evaluate them, improvements are limited. Future research directions could explore ways to reduce reliance on the critic or extend GFP to settings with weaker or sparse reward signals.

## ACKNOWLEDGMENTS

This work has received support from the French government, managed by the National Research Agency, under the France 2030 program with the references Organic Robotics Program (PEPR O2R) and "PR[AI]RIE-PSAI" (ANR-23-IACL-0008). This research was funded, in part, by l'Agence Nationale de la Recherche (ANR), projects NIMBLE project (ANR-22-CE33-0008), RODEO (ANR-24-CE23-5886), PEPR O2R - AS2 (ANR-22-EXOD-0006), and PEPR O2R – PI3 ASSISTMOV (ANR-22-EXOD-0004). The European Union also supported this work through the ARTIFACT project (GA no. 101165695) and the AGIMUS project (GA no. 101070165). The Paris Île-de-France Région also supported this work in the frame of the DIM AI4IDF. The authors gratefully acknowledge the support and resources provided by the CLEPS infrastructure at Inria Paris. This work was performed using HPC resources from the GENCI-IDRIS Jean-Zay cluster (Grant 2024-AD010616763). Views and opinions expressed are those of the author(s) only and do not necessarily reflect those of the funding agencies.

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

APPENDIX CONTENTS

*Note on LLM usage:* In this work, we only used LLMs for grammar and spelling corrections.

## A   IMPLEMENTATION DETAILS

**Network architectures.** All critic, actor, and flow neural networks use $[512, 512, 512, 512]$ multi-layer perceptrons with GeLU activations. Layer normalization is applied to the critic network.

**Flow matching.** The number of Euler steps, $M$, used in Algo. 1, Line 1, is fixed to 10 for both GFP and FQL, except on the humanoidmaze-large-navigate environment, where we set $M$ to 30. For GFP, we employ a sinusoidal position embedding of the flow step $t$, with an embedding size of 64.

**Doubled Q-learning.** Following standard practice, two separate critic networks are trained and then aggregated to compute action values, either by taking the mean or minimum (Fujimoto et al., 2018). As detailed in Sec. 4.3, we find that the aggregation function has a significant impact on performance for specific tasks. Specifically, by reevaluating ReBRAC on OGBench using the same aggregation function as GFP and FQL, we achieved substantial performance improvements.

**Minibatch size.** We use a minibatch size of $B = 256$ across most experiments, except on the most challenging tasks, where we evaluate each method with both $B = 256$ and $B = 1024$. The humanoidmaze-large-navigate environment is the only task where methods benefit from different batch sizes: GFP performs best with $B = 1024$, while other methods work better with $B = 256$. Note that on this task, using $\gamma = 0.999$ substantially improves the performance of ReBRAC and FQL compared to previously reported results. For Minari Gym-Mujoco, we use $B = 1024$ following the recommendation in Tarasov et al. (2023) for D4RL Gym-Mujoco.

**Training and evaluation.** To ensure a fair comparison with FQL, we use identical training durations: 1 M gradient steps on OGBench state-based environments, and 500 K steps on D4RL and OGBench pixel-based tasks. For Minari, we adopt 1 M steps following standard practices. Evaluation differs according to the benchmark: D4RL and Minari scores are computed at the end of training, while OGBench scores are averaged over the final three checkpoints (800K, 900K, and 1M steps for state-based, 300K, 400K and 500K steps for pixel-based) following their official evaluation protocol. All results are reported across 8 random seeds that were not used during the hyperparameter tuning process, and each evaluation is done over 100 episodes. Except for OGBench pixel-based tasks, where scores are computed using 50 episodes and 4 seeds, due to high computational cost, following Park et al. (2025).

**Minari MuJoCo reference scores.** Since reference scores for the MuJoCo locomotion tasks are not yet available in Minari (Younis et al., 2024), we use the reference scores reported in its predecessor D4RL (Fu et al., 2020).

**Parameter search methodology.** Tab. 4 summarizes the set of hyperparameter values shared across environments and methods. For task and method specific hyperparameters, our search follows a systematic approach. First, we conduct a logarithmic sweep over the BC coefficient $\alpha$, which is the main parameter for all methods. For ReBRAC, we sweep over the actor coefficient $\alpha_1$ while keeping

the critic coefficient $\alpha_2$ fixed at $0.01$. Then, we sweep over $\alpha_2$ after selecting the optimal $\alpha_1$. For GFP, we fix $\eta = 10^{-3}$ first and then sweep over $\eta$ once $\alpha$ is chosen. Sec. B.5 studies how sensitive GFP is to $\eta$ and $\alpha$, and justifies first searching $\alpha$ before adjusting $\eta$. Hyperparameter search uses four seeds, separate from the eight seeds used for final evaluation. On OGBench, hyperparameters are shared across the five tasks within each environment.

Table 4: **Summary of shared hyperparameters used across all methods and benchmark evaluations.** Environment-specific variations are indicated where applicable.

| Hyperparameter | Value |
|---|---|
| Learning rate | 0.0003 |
| Gradient steps | 1,000,000 (OGBench, Minari), 500,000 (D4RL) |
| Minibatch size | 256 (default), 1024 (Minari Gym-Mujoco), OGBench Tab. 5 |
| Discount factor | 0.99 (D4RL, Minari), OGBench Tab. 5 |
| Euler integration steps | 10 (default), 30 (humanoidmaze-large) |
| Critic aggregation function | mean (default), min (D4RL-antmaze, OGBench-antmaze) |
| Critic target network smoothing coefficient | 0.005 |
| Bellman target | $y = r + \gamma Q(s', a')$ (default), $y^{\text{VaBC}}$ (cube, humanoidmaze-medium) |

Table 5: **Discount factor and minibatch size for OGBench environments.** The asterisk * in some discount factors indicates cases where we modified the discount factor used in prior work, which led to significant performance improvements for the corresponding methods.

| Task Category | Discount factor $\gamma$ | Minibatch size |
|---|---|---|
| antmaze-large-navigate-singletask (5 tasks) | 0.995* | 256 |
| antmaze-large-stitch-singletask (5 tasks) | 0.995 (except for FQL, 0.99) | 256 |
| antmaze-large-explore-singletask (5 tasks) | 0.995 | 1024 |
| antmaze-giant-navigate-singletask (5 tasks) | 0.995 | 1024 |
| humanoidmaze-medium-navigate-singletask (5 tasks) | 0.995 | 256 |
| humanoidmaze-medium-stitch-singletask (5 tasks) | 0.999 | 256 |
| humanoidmaze-large-navigate-singletask (5 tasks) | 0.999* | 256 (except for GFP, 1024) |
| antsoccer-arena-navigate-singletask (5 tasks) | 0.99 | 256 |
| antsoccer-arena-stitch-singletask (5 tasks) | 0.99 | 256 |
| cube-single-play-singletask (5 tasks) | 0.99 | 256 |
| cube-single-noisy-singletask (5 tasks) | 0.99 | 256 |
| cube-double-play-singletask (5 tasks) | 0.99 | 256 |
| cube-double-noisy-singletask (5 tasks) | 0.99 | 256 |
| cube-triple-play-singletask (5 tasks) | 0.99 | 1024 |
| cube-triple-noisy-singletask (5 tasks) | 0.99 | 1024 |
| puzzle-3×3-play-singletask (5 tasks) | 0.99 | 256 |
| puzzle-4×4-play-singletask (5 tasks) | 0.99 | 256 |
| puzzle-4×4-noisy-singletask (5 tasks) | 0.99 | 256 |
| scene-play-singletask | 0.99 | 256 |
| scene-noisy-singletask | 0.99 | 256 |
| visual tasks | 0.99 | 256 |

# B ADDITIONAL EXPERIMENTS

## B.1 MODIFIED BELLMAN TARGET

As described in Sec. 3, we propose a variant of the Bellman target, $y^{\text{VaBC}}$ (Eq. 7), that leverages the VaBC policy. Tab. 6 presents experimental results showing average scores over 8 seeds for the selected hyperparameters, with "$\sim$" indicating configurations that were tested but not ultimately chosen. These experiments show that this modified target provides improvements in the cube and humanoidmaze-medium environments, and constant or slightly degraded results on other environments, hence $y^{\text{VaBC}}$ is used only for the former ones, as reported in Tab. 4.

*Implementation Note on Eq. 7*: After finishing the experiments, we realized that in our implementation of the modified $y^{\text{VaBC}}$ Bellman target, we were computing $Q_{\bar{\phi}}(s', \mu_\omega(s, z))$ instead of $Q_{\bar{\phi}}(s', \mu_\omega(s', z))$. In our public release of the code, we kept this behavior by default to match the code used in our experiments, but we added an option to use $Q_{\bar{\phi}}(s', \mu_\omega(s', z))$, for future experiments.

Table 6: **Comparison of the modified Bellman target for GFP.**

| Task Category | Standard target y | Modified $y^{\text{VaBC}}$ Eq. 7 |
|---|---|---|
| cube-double-play (5 tasks) | $\sim 28$ | $47.2 \pm_{1.6}$ |
| cube-double-noisy (5 tasks) | $\sim 46$ | $63.1 \pm_{3.3}$ |
| humanoidmaze-medium-navigate (5 tasks) | $\sim 64$ | $72.0 \pm_{2.8}$ |
| humanoidmaze-medium-stitch (5 tasks) | $\sim 59$ | $66.2 \pm_{5.7}$ |
| puzzle-4x4-play (5 tasks) | $26.1 \pm_{2.1}$ | $\sim 22$ |
| scene-play (5 tasks) | $53.5 \pm_{2.9}$ | $\sim 50$ |

## B.2 Advantage weight similar to AWR

As explained at the end of Sec. 3, our $g_\eta$ is a soft-max weight comparing $a$ to $a^{\pi_\theta}$, an action sampled from the actor, but one could alternatively use an advantage weighted term similar to AWR (Peng et al., 2019), but using the actor to compute a baseline:

$$g_\eta^{\text{AWR}}(s, a) := \exp\left(\tfrac{\lambda}{\eta}\big(Q_\phi(s,a) - Q_\phi(s, \mu_\theta(s,z))\big)\right) \tag{12}$$

Tab. 7 reports results when using $g_\eta^{\text{AWR}}$ instead of $g_\eta$, over 65 tasks, with $\alpha$ and $\eta$ tuned accordingly, Tab. 13. The modified Bellman target $y^{\text{VaBC}}$ has also been tested for this variant, as in Sec. B.1, and was found to be beneficial for the cube, humanoidmaze, and puzzle environments.

$g_\eta^{\text{AWR}}$ achieves slightly better performance on standard tasks, but our main $g_\eta$ scales better on more challenging ones, e.g., humanoid environments and cube-triple ones. Important note: GFP-AWR only changes the weighting function, keeping the general structure presented in Fig 1 and Fig 5f. In particular, it does not mean first using AWR to learn a weighted-BC policy (like FAWAC, see Fig 5d) and then using it to regularize the actor. GFP-AWR preserves the bidirectional guidance, a core contribution of GFP (see the arrows in Fig 5f). As an alternative, one could also learn a value-function network $V(s) = \mathbb{E}_{s;a\sim\pi_\theta(\cdot|s)}(Q(s,a))$ and use it instead of the stochastic baseline $Q(s, \mu_\theta(s,z))$ in the weighting functions.

Table 7: **Comparison with an AWR guiding function**

| Task Category | GFP-$g_\eta$ (default) | GFP-AWR |
|---|---|---|
| antmaze-large-stitch (5 tasks) | $\mathbf{68.9} \pm_{0.8}$ | $\mathbf{69.8} \pm_{1.9}$ |
| antmaze-giant-navigate (5 tasks) | $\mathbf{27.9} \pm_{8.5}$ | $\mathbf{29.1} \pm_{8.4}$ |
| humanoidmaze-medium-navigate (5 tasks) | $\mathbf{72.0} \pm_{2.8}$ | $60.9 \pm_{3.4}$ |
| humanoidmaze-medium-stitch (5 tasks) | $\mathbf{66.2} \pm_{5.7}$ | $\mathbf{64.3} \pm_{8.5}$ |
| humanoidmaze-large-navigate (5 tasks) | $\mathbf{17.0} \pm_{9.6}$ | $14.3 \pm_{3.5}$ |
| antsoccer-arena-stitch (5 tasks) | $\mathbf{30.5} \pm_{2.2}$ | $29.5 \pm_{2.6}$ |
| cube-double-play (5 tasks) | $47.2 \pm_{1.6}$ | $\mathbf{50.4} \pm_{3.5}$ |
| cube-double-noisy (5 tasks) | $\mathbf{63.1} \pm_{3.3}$ | $55.7 \pm_{3.4}$ |
| cube-triple-play(5 tasks) | $\mathbf{15.9} \pm_{2.0}$ | $9.9 \pm_{2.9}$ |
| cube-triple-noisy (5 tasks) | $\mathbf{24.5} \pm_{2.8}$ | $11.2 \pm_{1.4}$ |
| puzzle-3x3-play (5 tasks) | $23.1 \pm_{2.2}$ | $\mathbf{28.3} \pm_{1.0}$ |
| puzzle-4x4-play (5 tasks) | $\mathbf{18.8} \pm_{1.7}$ | $16.5 \pm_{1.7}$ |
| puzzle-4x4-noisy (5 tasks) | $\mathbf{26.1} \pm_{2.1}$ | $\mathbf{27.1} \pm_{1.7}$ |
| **Average** (65 tasks) | $\mathbf{38.6}$ | $35.9$ |

## B.3 Modified guiding function to integrate $a^{\pi_\omega}$

Our guiding function $g_\eta$, defined in Eq. 10, compares a dataset action $a$ to $a^{\pi_\theta}$, an action sampled from the actor. Since we also sample actions using the VaBC flow policy (Algo. 1, Line 10), we can alternatively use $a^{\pi_\omega}$ for a more conservative guiding function, leading to a modified guided function:

$$g_\eta^{min}(s,a) = \frac{\exp\left(\tfrac{\lambda}{\eta}Q_\phi(s,a)\right)}{\exp\left(\tfrac{\lambda}{\eta}Q_\phi(s,a)\right) + \exp\left(\tfrac{\lambda}{\eta}\ \min\left(Q_\phi(s, a^{\pi_\theta}), Q_\phi(s, a^{\pi_\omega})\right)\right)} \tag{13}$$

This modified guiding function is more conservative because it only filters out action $a$ when both policies produce more valuable alternatives. Initially, we employed $g_\eta^{min}(s,a)$ for evaluation on

OGBench antmaze, humanoidmaze, and antsoccer environments. However, we found no significant performance difference, so all remaining tasks use the standard $g_\eta$ defined in Eq. 10.

## B.4 TEMPERATURE ANALYSIS

To complete the analysis presented in Sec. 4.2, we conducted experiments on two additional tasks beyond those shown in Fig. 4. We tested humanoidmaze-medium-stitch as a locomotion task and cube-triple-play as a non-noisy manipulation task. The results of this extended temperature analysis are reported in Fig. 6, illustrating how temperature controls the sharpness of value-guided filtering.

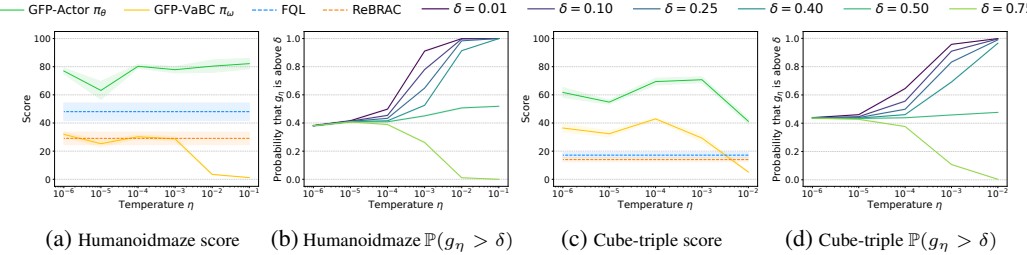

Figure 6: **Temperature analysis on two additional very challenging tasks.** Plots (a) and (c), performance scores across temperature values $\eta$ for our GFP method (Actor $\pi_\theta$ and VaBC $\pi_\omega$) compared to baselines (FQL, ReBRAC) on humanoidmaze-medium-stitch-task1 and cube-triple-play-task1. Plots (b) and (d), probability that the guidance term $g_\eta$ is above different thresholds $\delta$ as a function of temperature, illustrating how temperature controls the sharpness of value-guided filtering.

## B.5 SENSITIVITY ANALYSIS TO $\eta$ AND $\alpha$

To study how sensitive GFP is to the temperature $\eta$ and the BC coefficient $\alpha$, we evaluated 8 variations around task-specific hyperparameters $(\alpha^*, \eta^*)$ on 12 tasks, with comparison to FQL sensitivity to $\alpha$. These experiments reveal that $\alpha$ is the most important hyperparameter of GFP, as it is for most offline RL methods (Tarasov et al., 2023; Park et al., 2024b), and that GFP is less sensitive to precise $\eta$ tuning. It justifies our methodology for hyperparameter search, by first fixing $\eta = 10^{-3}$, and then sweeping over $\eta$ only once $\alpha$ is chosen. From the results averaged on 12 tasks, it can be observed that $\alpha$ and $\eta$ are partially correlated: when increasing $\alpha$ (i.e. stronger regularization), it is better to decrease $\eta$ (stronger filtering).

## C COMPLETE RESULTS OVER 144 TASKS

This section presents the comprehensive experimental results across all 144 tasks from OGBench (Tabs. 8 and 9), D4RL (Tab. 11), and Minari (Tab. 12) benchmarks. Tab. 10 covers comparison against 10 prior works over 50 tasks. All results are averaged over 8 seeds, except for pixel-based tasks. The evaluation encompasses approximately 15,000 individual training runs, providing detailed performance comparisons across diverse offline RL scenarios, including navigation, manipulation, and locomotion tasks. The hyperparameters used are stated in Tabs. 14 and 13.

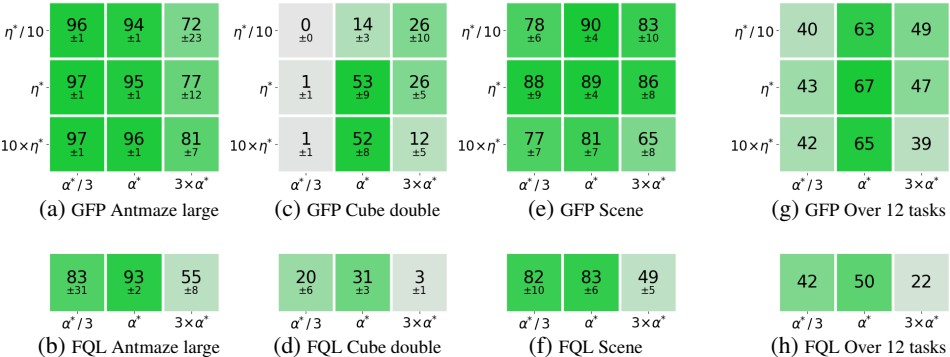

Figure 7: **Sensitivity analysis to the BC coefficient $\alpha$, and to the temperature $\eta$, in log-scale.** Evaluates the sensitivity by testing variations around task-specific hyperparameters $(\alpha^*, \eta^*)$ reported Tab. 13, using 8 seeds. Plots (a), (c) and (e), report GFP's sensitivity to $\alpha^*$ and $\eta^*$, on antmaze-large-navigate-task1, cube-double-play-task2, and scene-play-task2, respectively. These three tasks were chosen following FQL (Park et al., 2025) ablation study on $\alpha$ (FQL page 15). Plots (b), (d) and (f), report the sensitivity of FQL with respect to $\alpha$ (reevaluated as FQL's authors did not share exact numbers). Plots (g) and (h) aggregate the analysis over 12 tasks (cube single and double, play and noisy; scene play and noisy; antmaze navigate large and giant; humanoidmaze medium navigate and stitch, where for each environment we use the default task). These experiments reveal that $\alpha$ is the most important hyperparameter of GFP, as it is for most offline RL methods (Tarasov et al., 2023; Park et al., 2024b), and that GFP is less sensitive to precise $\eta$ tuning.

Table 8: **Offline RL full results on OGBench, part 1/2.** Models were trained with 8 random seeds and evaluated over 100 episodes, following the setup of prior work (Park et al., 2024a; 2025). Scores are averaged across seeds; values within 95% of the best performance are shown in bold, while *italics* indicate scores reported from prior work (Park et al., 2025). GFP actor $\pi_\theta$ is our primary policy, while GFP VaBC $\pi_\omega$ is reported as a byproduct of training. The $\pm$ symbol denotes the standard deviation over seeds. See Tab 9 for the other tasks.

| Task | Offline RL algorithms | | | | |
|---|---|---|---|---|---|
| | IQL | ReBRAC | FQL | **GFP** actor $\pi_\theta$ | **GFP** VaBC $\pi_\omega$ |
| antmaze-large-navigate-singletask-task1-v0 | *48 ± 9* | **97.7 ± 0.5** | 92.6 ± 1.8 | **95.4 ± 0.8** | 92.0 ± 2.3 |
| antmaze-large-navigate-singletask-task2-v0 | *42 ± 6* | **92.2 ± 1.6** | 80.0 ± 10.4 | **92.2 ± 3.0** | **88.5 ± 1.6** |
| antmaze-large-navigate-singletask-task3-v0 | *72 ± 7* | **98.5 ± 1.8** | 93.5 ± 1.6 | **95.6 ± 2.7** | **94.4 ± 3.0** |
| antmaze-large-navigate-singletask-task4-v0 | *51 ± 9* | **94.4 ± 0.9** | 82.3 ± 15.5 | **90.6 ± 2.6** | 85.6 ± 3.7 |
| antmaze-large-navigate-singletask-task5-v0 | *54 ± 22* | **96.5 ± 0.9** | **92.3 ± 1.5** | **95.0 ± 1.3** | 89.8 ± 1.7 |
| antmaze-large-stitch-singletask-task1-v0 | 28.2 ± 7.8 | **88.4 ± 16.2** | 71.8 ± 28.4 | **90.3 ± 2.1** | 85.0 ± 9.7 |
| antmaze-large-stitch-singletask-task2-v0 | 5.5 ± 4.1 | **85.2 ± 5.9** | 25.4 ± 25.3 | **82.9 ± 2.3** | 69.5 ± 7.2 |
| antmaze-large-stitch-singletask-task3-v0 | 83.4 ± 2.5 | **98.0 ± 0.8** | 88.4 ± 3.5 | **93.2 ± 3.4** | 90.2 ± 1.5 |
| antmaze-large-stitch-singletask-task4-v0 | 8.8 ± 2.9 | **79.4 ± 29.9** | 23.7 ± 24.6 | 0.2 ± 0.6 | 1.2 ± 3.1 |
| antmaze-large-stitch-singletask-task5-v0 | 26.2 ± 14.9 | **95.1 ± 1.7** | 81.5 ± 7.2 | 77.9 ± 7.1 | 42.2 ± 8.7 |
| antmaze-large-explore-singletask-task1-v0 | 0.4 ± 1.0 | **91.8 ± 6.1** | **91.0 ± 10.9** | **92.8 ± 4.1** | **92.3 ± 1.0** |
| antmaze-large-explore-singletask-task2-v0 | 0.0 ± 0.0 | 86.4 ± 5.2 | **90.8 ± 3.4** | **88.5 ± 3.6** | 84.1 ± 3.1 |
| antmaze-large-explore-singletask-task3-v0 | 54.8 ± 7.7 | **99.1 ± 0.6** | 97.5 ± 0.9 | **98.0 ± 0.5** | 92.8 ± 1.1 |
| antmaze-large-explore-singletask-task4-v0 | 9.2 ± 4.1 | 54.4 ± 30.5 | **89.0 ± 2.9** | 87.1 ± 2.3 | 87.0 ± 2.8 |
| antmaze-large-explore-singletask-task5-v0 | 0.0 ± 0.1 | 81.8 ± 23.7 | 71.1 ± 32.0 | **93.1 ± 2.1** | **90.2 ± 9.7** |
| antmaze-giant-navigate-singletask-task1-v0 | *0 ± 0* | **17.5 ± 3.8** | 0.8 ± 2.1 | 12.6 ± 15.1 | 0.1 ± 0.1 |
| antmaze-giant-navigate-singletask-task2-v0 | *1 ± 1* | 44.9 ± 6.7 | 23.2 ± 15.3 | **52.2 ± 26.5** | 1.2 ± 0.7 |
| antmaze-giant-navigate-singletask-task3-v0 | *0 ± 0* | 2.5 ± 1.3 | 0.9 ± 1.1 | **13.7 ± 10.2** | 0.2 ± 0.2 |
| antmaze-giant-navigate-singletask-task4-v0 | *0 ± 0* | **20.0 ± 20.0** | 9.9 ± 10.8 | 17.8 ± 19.9 | 0.4 ± 0.3 |
| antmaze-giant-navigate-singletask-task5-v0 | 19 ± 7 | **81.4 ± 6.1** | 46.9 ± 25.5 | 43.2 ± 35.7 | 2.0 ± 0.7 |
| humanoidmaze-medium-navigate-singletask-task1-v0 | *32 ± 7* | 34.1 ± 16.4 | *19 ± 12* | **83.5 ± 3.7** | 28.8 ± 6.5 |
| humanoidmaze-medium-navigate-singletask-task2-v0 | *41 ± 9* | 75.8 ± 29. | *94 ± 3* | **91.2 ± 6.3** | 60.2 ± 9.4 |
| humanoidmaze-medium-navigate-singletask-task3-v0 | *25 ± 5* | 69.5 ± 25.7 | *74 ± 18* | **86.3 ± 10.7** | 28.9 ± 4.1 |
| humanoidmaze-medium-navigate-singletask-task4-v0 | *0 ± 1* | **19.5 ± 17.5** | *3 ± 4* | 3.0 ± 6.6 | 1.3 ± 2.0 |
| humanoidmaze-medium-navigate-singletask-task5-v0 | *66 ± 4* | **97.0 ± 0.9** | *97 ± 2* | **95.8 ± 1.3** | 60.1 ± 6.1 |
| humanoidmaze-medium-stitch-singletask-task1-v0 | 26.4 ± 3.0 | 29.1 ± 18.3 | 48.0 ± 25.4 | **77.9 ± 9.2** | 28.7 ± 6.5 |
| humanoidmaze-medium-stitch-singletask-task2-v0 | 27.9 ± 9.9 | **94.4 ± 1.9** | 87.5 ± 3.7 | **95.2 ± 1.8** | 49.2 ± 6.3 |
| humanoidmaze-medium-stitch-singletask-task3-v0 | 30.0 ± 4.4 | 56.6 ± 24.5 | **85.4 ± 17.8** | 55.2 ± 33.9 | 44.3 ± 3.2 |
| humanoidmaze-medium-stitch-singletask-task4-v0 | 3.7 ± 1.6 | **33.1 ± 28.4** | 0.8 ± 0.8 | 3.6 ± 9.5 | 14.9 ± 9.0 |
| humanoidmaze-medium-stitch-singletask-task5-v0 | 48.5 ± 4.6 | 92.5 ± 3.1 | 94.1 ± 2.5 | **98.9 ± 0.5** | 60.2 ± 7.4 |
| humanoidmaze-large-navigate-singletask-task1-v0 | *3 ± 1* | 27.8 ± 13.4 | 19.8 ± 13.1 | **57.2 ± 23.7** | 4.5 ± 2.9 |
| humanoidmaze-large-navigate-singletask-task2-v0 | *0 ± 0* | **0.5 ± 0.7** | 0.0 ± 0.1 | 0.1 ± 0.2 | 0.0 ± 0.0 |
| humanoidmaze-large-navigate-singletask-task3-v0 | *7 ± 3* | **25.9 ± 8.5** | 8.8 ± 4.0 | 14.6 ± 16.2 | 5.5 ± 2.8 |
| humanoidmaze-large-navigate-singletask-task4-v0 | *1 ± 0* | **8.3 ± 14.4** | 1.6 ± 1.8 | 3.7 ± 4.1 | 0.7 ± 0.5 |
| humanoidmaze-large-navigate-singletask-task5-v0 | *1 ± 1* | 1.9 ± 1.5 | 2.2 ± 3.9 | **13.1 ± 15.5** | 1.5 ± 0.9 |
| antsoccer-arena-navigate-singletask-task1-v0 | *14 ± 5* | 62.1 ± 3.6 | *77 ± 4* | **77.0 ± 1.7** | 17.0 ± 2.8 |
| antsoccer-arena-navigate-singletask-task2-v0 | *17 ± 7* | 78.5 ± 2.8 | *88 ± 3* | **91.2 ± 2.2** | 16.8 ± 2.6 |
| antsoccer-arena-navigate-singletask-task3-v0 | *6 ± 4* | 55.5 ± 1.7 | *61 ± 6* | 51.9 ± 4.9 | 7.8 ± 2.9 |
| antsoccer-arena-navigate-singletask-task4-v0 | *3 ± 2* | 34.8 ± 5.0 | *39 ± 6* | **40.2 ± 4.2** | 5.2 ± 2.1 |
| antsoccer-arena-navigate-singletask-task5-v0 | *2 ± 2* | **48.5 ± 6.1** | 36 ± 9 | 29.1 ± 8.9 | 4.6 ± 2.1 |
| antsoccer-arena-stitch-singletask-task1-v0 | 5.3 ± 3.3 | 44.6 ± 5.0 | **53.4 ± 3.5** | 51.8 ± 3.5 | 3.0 ± 1.2 |
| antsoccer-arena-stitch-singletask-task2-v0 | 5.6 ± 1.9 | 30.0 ± 6.0 | 49.1 ± 8.1 | **53.0 ± 7.6** | 3.0 ± 1.4 |
| antsoccer-arena-stitch-singletask-task3-v0 | 1.3 ± 1.7 | 15.9 ± 2.4 | **19.3 ± 2.7** | 18.7 ± 1.4 | 0.4 ± 0.3 |
| antsoccer-arena-stitch-singletask-task4-v0 | 0.4 ± 0.5 | 14.8 ± 4.2 | 20.0 ± 6.4 | **26.1 ± 4.7** | 0.1 ± 0.2 |
| antsoccer-arena-stitch-singletask-task5-v0 | 1.3 ± 1.8 | **4.8 ± 1.6** | 1.2 ± 0.4 | 2.9 ± 2.9 | 0.6 ± 0.4 |
| cube-single-play-singletask-task1-v0 | *88 ± 3* | 89 ± 5 | *97 ± 2* | **99.1 ± 0.4** | 42.1 ± 5.9 |
| cube-single-play-singletask-task2-v0 | *85 ± 8* | 92 ± 4 | *97 ± 2* | **99.4 ± 0.7** | 38.8 ± 6.4 |
| cube-single-play-singletask-task3-v0 | *91 ± 5* | 93 ± 3 | *98 ± 2* | **99.4 ± 0.5** | 48.5 ± 9.1 |
| cube-single-play-singletask-task4-v0 | *73 ± 6* | 92 ± 3 | *94 ± 3* | **99.1 ± 0.7** | 32.8 ± 9.3 |
| cube-single-play-singletask-task5-v0 | *78 ± 9* | 87 ± 8 | *93 ± 3* | **97.0 ± 1.6** | 36.3 ± 5.5 |
| cube-single-noisy-singletask-task1-v0 | 52.3 ± 7.2 | 99.2 ± 1.1 | **100.0 ± 0.0** | **100.0 ± 0.0** | **99.9 ± 0.2** |
| cube-single-noisy-singletask-task2-v0 | 55.3 ± 8.0 | 96.0 ± 3.5 | **100.0 ± 0.1** | **100.0 ± 0.1** | **99.9 ± 0.2** |
| cube-single-noisy-singletask-task3-v0 | 34.3 ± 8.1 | 97.4 ± 1.6 | **100.0 ± 0.0** | **100.0 ± 0.0** | **100.0 ± 0.0** |
| cube-single-noisy-singletask-task4-v0 | 63.2 ± 7.5 | 99.7 ± 0.5 | **100.0 ± 0.1** | **100.0 ± 0.0** | **99.9 ± 0.1** |
| cube-single-noisy-singletask-task5-v0 | 60.9 ± 11.7 | **99.8 ± 0.2** | **100.0 ± 0.1** | **99.9 ± 0.2** | **99.8 ± 0.3** |

Table 9: **Offline RL full results on OGBench, part 2/2.** Models were trained with 8 random seeds and evaluated over 100 episodes, following the setup of prior work (Park et al., 2024a; 2025). Scores are averaged across seeds; values within 95% of the best performance are shown in bold, while *italics* indicate scores reported from prior work (Park et al., 2025). GFP actor $\pi_\theta$ is our primary policy, while GFP VaBC $\pi_\omega$ is reported as a byproduct of training. The $\pm$ symbol denotes the standard deviation over seeds. See Tab 8 for the other tasks.

| Task | Offline RL algorithms | | | | |
| --- | --- | --- | --- | --- | --- |
| | IQL | ReBRAC | FQL | **GFP** actor $\pi_\omega$ | **GFP** VaBC $\pi_\theta$ |
| cube-double-play-singletask-task1-v0 | *27* $\pm$ *5* | 43.0 $\pm$ 8.9 | *61* $\pm$ *9* | **76.1** $\pm$ 4.6 | 28.5 $\pm$ 5.0 |
| cube-double-play-singletask-task2-v0 | *1* $\pm$ *1* | 16.2 $\pm$ 5.0 | *36* $\pm$ *6* | **53.3** $\pm$ 8.9 | 1.8 $\pm$ 1.0 |
| cube-double-play-singletask-task3-v0 | *0* $\pm$ *0* | 1.3 $\pm$ 0.4 | *22* $\pm$ *5* | **43.3** $\pm$ 8.9 | 0.4 $\pm$ 0.4 |
| cube-double-play-singletask-task4-v0 | *0* $\pm$ *0* | 0.4 $\pm$ 0.3 | *5* $\pm$ *2* | **7.1** $\pm$ 3.1 | 0.8 $\pm$ 0.6 |
| cube-double-play-singletask-task5-v0 | *4* $\pm$ *3* | 2.0 $\pm$ 1.0 | *19* $\pm$ *10* | **56.3** $\pm$ 11.3 | 0.7 $\pm$ 0.6 |
| cube-double-noisy-singletask-task1-v0 | 20.8 $\pm$ 3.4 | 51.3 $\pm$ 9.5 | 77.1 $\pm$ 8.0 | **89.5** $\pm$ 4.5 | 32.2 $\pm$ 3.9 |
| cube-double-noisy-singletask-task2-v0 | 0.0 $\pm$ 0.1 | 21.1 $\pm$ 4.3 | 43.1 $\pm$ 10.5 | **75.7** $\pm$ 7.6 | 5.8 $\pm$ 1.3 |
| cube-double-noisy-singletask-task3-v0 | 0.8 $\pm$ 1.0 | 8.0 $\pm$ 3.4 | 26.3 $\pm$ 5.8 | **75.0** $\pm$ 4.4 | 3.2 $\pm$ 1.2 |
| cube-double-noisy-singletask-task4-v0 | 0.2 $\pm$ 0.2 | 6.5 $\pm$ 1.8 | 15.5 $\pm$ 3.9 | **41.8** $\pm$ 4.6 | 1.6 $\pm$ 0.9 |
| cube-double-noisy-singletask-task5-v0 | 0.5 $\pm$ 0.5 | 11.2 $\pm$ 3.3 | 29.0 $\pm$ 7.9 | **33.4** $\pm$ 7.6 | 3.9 $\pm$ 1.5 |
| cube-triple-play-singletask-task1-v0 | 0.4 $\pm$ 0.3 | 14.0 $\pm$ 5.8 | 17.2 $\pm$ 7.3 | **54.8** $\pm$ 6.2 | 32.4 $\pm$ 8.4 |
| cube-triple-play-singletask-task2-v0 | 0.0 $\pm$ 0.0 | 0.1 $\pm$ 0.1 | 0.8 $\pm$ 0.2 | **6.6** $\pm$ 6.3 | 0.6 $\pm$ 0.7 |
| cube-triple-play-singletask-task3-v0 | 1.3 $\pm$ 0.6 | 0.3 $\pm$ 0.3 | 1.3 $\pm$ 0.6 | **14.9** $\pm$ 9.9 | 3.2 $\pm$ 2.1 |
| cube-triple-play-ingletask-task4-v0 | 0.0 $\pm$ 0.0 | 0.0 $\pm$ 0.0 | 0.3 $\pm$ 0.4 | **2.5** $\pm$ 1.7 | 0.8 $\pm$ 0.2 |
| cube-triple-play-singletask-task5-v0 | 0.1 $\pm$ 0.2 | 0.3 $\pm$ 0.5 | 0.1 $\pm$ 0.2 | **0.6** $\pm$ 0.5 | **1.0** $\pm$ 0.9 |
| cube-triple-noisy-singletask-task1-v0 | 24.0 $\pm$ 6.0 | 25.3 $\pm$ 13.9 | 17.5 $\pm$ 8.0 | **90.7** $\pm$ 5.2 | 41.0 $\pm$ 6.6 |
| cube-triple-noisy-singletask-task2-v0 | 0.0 $\pm$ 0.1 | 0.4 $\pm$ 0.4 | 0.1 $\pm$ 0.2 | **8.9** $\pm$ 3.4 | 0.3 $\pm$ 0.2 |
| cube-triple-noisy-singletask-task3-v0 | 0.0 $\pm$ 0.0 | 0.1 $\pm$ 0.2 | 0.0 $\pm$ 0.0 | **11.8** $\pm$ 7.5 | 0.8 $\pm$ 0.7 |
| cube-triple-noisy-ingletask-task4-v0 | 0.0 $\pm$ 0.0 | 0.0 $\pm$ 0.0 | 0.0 $\pm$ 0.1 | **10.8** $\pm$ 6.2 | 0.8 $\pm$ 0.5 |
| cube-triple-noisy-singletask-task5-v0 | 0.0 $\pm$ 0.0 | 0.0 $\pm$ 0.1 | 0.0 $\pm$ 0.0 | **0.1** $\pm$ 0.2 | **0.0** $\pm$ 0.0 |
| puzzle-3$\times$3-play-singletask-task1-v0 | *33* $\pm$ *6* | **97** $\pm$ *4* | *90* $\pm$ *4* | **94.8** $\pm$ 4.4 | 54.6 $\pm$ 7.4 |
| puzzle-3$\times$3-play-singletask-task2-v0 | *4* $\pm$ *3* | *1* $\pm$ *1* | **16** $\pm$ *5* | 0.3 $\pm$ 0.3 | 11.2 $\pm$ 2.4 |
| puzzle-3$\times$3-play-singletask-task3-v0 | *3* $\pm$ *2* | *3* $\pm$ *1* | **10** $\pm$ *3* | 0.9 $\pm$ 0.6 | **9.7** $\pm$ 1.5 |
| puzzle-3$\times$3-play-singletask-task4-v0 | *2* $\pm$ *1* | *2* $\pm$ *1* | **16** $\pm$ *5* | 5.4 $\pm$ 2.1 | 9.4 $\pm$ 3.1 |
| puzzle-3$\times$3-play-singletask-task5-v0 | *3* $\pm$ *2* | *5* $\pm$ *3* | **16** $\pm$ *3* | 14.1 $\pm$ 8.4 | 10.9 $\pm$ 4.8 |
| puzzle-4$\times$4-play-singletask-task1-v0 | *12* $\pm$ *2* | 45.4 $\pm$ 3.7 | *34* $\pm$ *8* | **50.0** $\pm$ 8.1 | 16.2 $\pm$ 3.3 |
| puzzle-4$\times$4-play-singletask-task2-v0 | *7* $\pm$ *4* | 2.7 $\pm$ 1.0 | **16** $\pm$ *5* | 9.9 $\pm$ 2.5 | 7.0 $\pm$ 1.8 |
| puzzle-4$\times$4-play-singletask-task3-v0 | *9* $\pm$ *3* | 27.8 $\pm$ 4.0 | *18* $\pm$ *5* | **46.2** $\pm$ 3.8 | 10.2 $\pm$ 2.9 |
| puzzle-4$\times$4-play-singletask-task4-v0 | *5* $\pm$ *2* | 9.1 $\pm$ 2.1 | *11* $\pm$ *3* | **17.2** $\pm$ 2.5 | 7.4 $\pm$ 1.9 |
| puzzle-4$\times$4-play-singletask-task5-v0 | *4* $\pm$ *1* | 0.8 $\pm$ 0.7 | **7** $\pm$ *3* | **7.3** $\pm$ 3.6 | **6.6** $\pm$ 1.8 |
| puzzle-4$\times$4-noisy-singletask-task1-v0 | 0.1 $\pm$ 0.1 | 3.9 $\pm$ 1.2 | **41.0** $\pm$ 3.8 | 38.5 $\pm$ 3.6 | **39.6** $\pm$ 4.8 |
| puzzle-4$\times$4-noisy-singletask-task2-v0 | 0.0 $\pm$ 0.1 | 0.4 $\pm$ 0.4 | **5.9** $\pm$ 1.7 | 0.7 $\pm$ 0.5 | 3.5 $\pm$ 1.1 |
| puzzle-4$\times$4-noisy-singletask-task3-v0 | 0.1 $\pm$ 0.2 | 0.9 $\pm$ 0.5 | 20.8 $\pm$ 2.7 | **51.1** $\pm$ 6.5 | 44.0 $\pm$ 4.7 |
| puzzle-4$\times$4-noisy-singletask-task4-v0 | 0.0 $\pm$ 0.0 | 0.4 $\pm$ 0.4 | **6.5** $\pm$ 1.6 | 3.0 $\pm$ 1.6 | **6.3** $\pm$ 2.2 |
| puzzle-4$\times$4-noisy-singletask-task5-v0 | 0.0 $\pm$ 0.0 | 0.0 $\pm$ 0.1 | **3.7** $\pm$ 1.7 | 0.7 $\pm$ 0.7 | **3.1** $\pm$ 2.0 |
| scene-play-singletask-task1-v0 | *94* $\pm$ *3* | **95** $\pm$ *2* | **100** $\pm$ *0* | **99.8** $\pm$ 0.4 | **99.8** $\pm$ 0.2 |
| scene-play-singletask-task2-v0 | *12* $\pm$ *3* | *50* $\pm$ *13* | *76* $\pm$ *9* | 89.0 $\pm$ 4.1 | **93.0** $\pm$ 5.1 |
| scene-play-singletask-task3-v0 | *32* $\pm$ *7* | *55* $\pm$ *16* | **98** $\pm$ *1* | 78.0 $\pm$ 13.2 | **93.5** $\pm$ 5.1 |
| scene-play-singletask-task4-v0 | *0* $\pm$ *0* | *3* $\pm$ *3* | **5** $\pm$ *1* | 0.6 $\pm$ 0.6 | 1.8 $\pm$ 1.3 |
| scene-play-singletask-task5-v0 | *0* $\pm$ *0* | *0* $\pm$ *0* | *0* $\pm$ *0* | **0.0** $\pm$ 0.0 | **0.0** $\pm$ 0.0 |
| scene-noisy-singletask-task1-v0 | 74.2 $\pm$ 5.4 | 94.8 $\pm$ 3.7 | **100.0** $\pm$ 0.0 | **99.9** $\pm$ 0.2 | **99.9** $\pm$ 0.2 |
| scene-noisy-singletask-task2-v0 | 0.1 $\pm$ 0.2 | 18.1 $\pm$ 5.9 | 87.4 $\pm$ 3.7 | 94.2 $\pm$ 2.0 | **97.4** $\pm$ 1.9 |
| scene-noisy-singletask-task3-v0 | 5.7 $\pm$ 1.1 | 81.1 $\pm$ 5.5 | 94.4 $\pm$ 3.7 | 93.3 $\pm$ 4.3 | **95.2** $\pm$ 3.2 |
| scene-noisy-singletask-task4-v0 | 0.0 $\pm$ 0.1 | 5.6 $\pm$ 3.4 | **14.8** $\pm$ 4.6 | 0.0 $\pm$ 0.0 | 0.1 $\pm$ 0.2 |
| scene-noisy-singletask-task5-v0 | 0.0 $\pm$ 0.0 | 0.0 $\pm$ 0.0 | 0.0 $\pm$ 0.0 | **0.0** $\pm$ 0.0 | **0.0** $\pm$ 0.0 |
| visual-cube-single-play-singletask-task1-v0[1] | *70* $\pm$ *12* | **83** $\pm$ *6* | **81** $\pm$ *12* | **82.3** $\pm$ 4.2 | $-$[1] |
| visual-cube-double-play-singletask-task1-v0[1] | **34** $\pm$ *23* | *4* $\pm$ *4* | *21* $\pm$ *11* | 19.7 $\pm$ 8.7 | $-$ |
| visual-scene-play-singletask-task1-v0[1] | *97* $\pm$ *2* | **98** $\pm$ *4* | **98** $\pm$ *3* | **99.3** $\pm$ 0.5 | $-$ |
| visual-puzzle-3x3-play-singletask-task1-v0[1] | *7* $\pm$ *15* | *88* $\pm$ *4* | **94** $\pm$ *1* | 92.8 $\pm$ 2.0 | $-$ |
| visual-puzzle-4x4-play-singletask-task1-v0[1] | *0* $\pm$ *0* | *26* $\pm$ *6* | **33** $\pm$ *6* | 19.7 $\pm$ 1.7 | $-$ |

[1] Following Park et al. (2025) to reduce the computational cost of pixel-based tasks, for these tasks we use 4 seeds and 50 episodes per seed; moreover, we chose not to evaluate the VaBC policy, as it is only a byproduct.

Table 10: **Offline RL Results on OGBench Tasks (only task evaluated in FQL):** Performance comparison across different offline RL algorithms. All prior work results are taken from Park et al. (2025) to compare a wide range of previous methods against GFP.

| Task | BC | IQL | ReBRAC | IDQL | SRPO | CAC | FAWAC | FBRAC | IFQL | FQL | GFP actor $\pi_\omega$ |
|---|---|---|---|---|---|---|---|---|---|---|---|
| *antmaze-large-navigate-singletask* | | | | | | | | | | | |
| task1-v0 | $0 \pm 0$ | $48 \pm 9$ | $91 \pm 10$ | $0 \pm 0$ | $0 \pm 0$ | $42 \pm 7$ | $1 \pm 1$ | $70 \pm 20$ | $24 \pm 17$ | $80 \pm 8$ | $\mathbf{95.4} \pm 0.8$ |
| task2-v0 | $6 \pm 3$ | $42 \pm 6$ | $88 \pm 4$ | $14 \pm 8$ | $4 \pm 4$ | $1 \pm 1$ | $0 \pm 1$ | $35 \pm 12$ | $8 \pm 3$ | $57 \pm 10$ | $\mathbf{92.2} \pm 3.0$ |
| task3-v0 | $29 \pm 5$ | $72 \pm 7$ | $51 \pm 18$ | $26 \pm 8$ | $3 \pm 2$ | $49 \pm 10$ | $12 \pm 4$ | $83 \pm 15$ | $52 \pm 17$ | $93 \pm 3$ | $\mathbf{95.6} \pm 2.7$ |
| task4-v0 | $8 \pm 3$ | $51 \pm 9$ | $84 \pm 7$ | $62 \pm 25$ | $45 \pm 19$ | $17 \pm 6$ | $10 \pm 3$ | $37 \pm 18$ | $18 \pm 8$ | $80 \pm 4$ | $\mathbf{90.6} \pm 2.6$ |
| task5-v0 | $10 \pm 3$ | $54 \pm 22$ | $90 \pm 2$ | $2 \pm 2$ | $1 \pm 1$ | $55 \pm 6$ | $9 \pm 5$ | $76 \pm 8$ | $38 \pm 18$ | $83 \pm 4$ | $\mathbf{95.0} \pm 1.3$ |
| *antmaze-giant-navigate-singletask-task1* | | | | | | | | | | | |
| task1-v0 | $0 \pm 0$ | $0 \pm 0$ | $\mathbf{27} \pm 22$ | $0 \pm 0$ | $0 \pm 0$ | $0 \pm 0$ | $0 \pm 0$ | $0 \pm 1$ | $0 \pm 0$ | $4 \pm 5$ | $12.6 \pm 15.1$ |
| task2-v0 | $0 \pm 0$ | $1 \pm 1$ | $16 \pm 17$ | $0 \pm 0$ | $0 \pm 0$ | $0 \pm 0$ | $0 \pm 0$ | $4 \pm 7$ | $0 \pm 0$ | $9 \pm 7$ | $\mathbf{52.2} \pm 26.5$ |
| task3-v0 | $0 \pm 0$ | $0 \pm 0$ | $\mathbf{34} \pm 22$ | $0 \pm 0$ | $0 \pm 0$ | $0 \pm 0$ | $0 \pm 0$ | $0 \pm 0$ | $0 \pm 0$ | $0 \pm 1$ | $13.7 \pm 10.2$ |
| task4-v0 | $0 \pm 0$ | $0 \pm 0$ | $5 \pm 12$ | $0 \pm 0$ | $0 \pm 0$ | $0 \pm 0$ | $0 \pm 0$ | $9 \pm 4$ | $0 \pm 0$ | $14 \pm 23$ | $\mathbf{17.8} \pm 19.9$ |
| task5-v0 | $1 \pm 1$ | $19 \pm 7$ | $\mathbf{49} \pm 22$ | $0 \pm 1$ | $0 \pm 0$ | $0 \pm 0$ | $0 \pm 0$ | $6 \pm 10$ | $13 \pm 9$ | $16 \pm 28$ | $43.2 \pm 35.7$ |
| *humanoidmaze-medium-navigate-singletask* | | | | | | | | | | | |
| task1-v0 | $1 \pm 0$ | $32 \pm 7$ | $16 \pm 9$ | $1 \pm 1$ | $0 \pm 0$ | $38 \pm 19$ | $6 \pm 2$ | $25 \pm 8$ | $69 \pm 19$ | $19 \pm 12$ | $\mathbf{83.5} \pm 3.7$ |
| task2-v0 | $1 \pm 0$ | $41 \pm 9$ | $18 \pm 16$ | $1 \pm 1$ | $1 \pm 1$ | $47 \pm 35$ | $40 \pm 2$ | $76 \pm 10$ | $85 \pm 11$ | $\mathbf{94} \pm 3$ | $91.2 \pm 6.3$ |
| task3-v0 | $6 \pm 2$ | $25 \pm 5$ | $36 \pm 13$ | $0 \pm 1$ | $2 \pm 1$ | $\mathbf{83} \pm 18$ | $19 \pm 2$ | $27 \pm 11$ | $49 \pm 49$ | $74 \pm 18$ | $86.3 \pm 10.7$ |
| task4-v0 | $0 \pm 0$ | $0 \pm 1$ | $15 \pm 16$ | $1 \pm 1$ | $1 \pm 1$ | $5 \pm 4$ | $1 \pm 1$ | $1 \pm 2$ | $1 \pm 1$ | $3 \pm 4$ | $3.0 \pm 6.6$ |
| task5-v0 | $2 \pm 1$ | $66 \pm 4$ | $24 \pm 20$ | $1 \pm 1$ | $3 \pm 3$ | $91 \pm 5$ | $31 \pm 7$ | $63 \pm 9$ | $\mathbf{98} \pm 2$ | $97 \pm 2$ | $95.8 \pm 1.3$ |
| *humanoidmaze-large-navigate-singletask* | | | | | | | | | | | |
| task1-v0 | $0 \pm 0$ | $3 \pm 1$ | $2 \pm 1$ | $0 \pm 0$ | $0 \pm 0$ | $1 \pm 1$ | $0 \pm 0$ | $0 \pm 1$ | $6 \pm 2$ | $7 \pm 6$ | $\mathbf{57.2} \pm 23.7$ |
| task2-v0 | $0 \pm 0$ | $0 \pm 0$ | $0 \pm 0$ | $0 \pm 0$ | $0 \pm 0$ | $0 \pm 0$ | $0 \pm 0$ | $0 \pm 0$ | $0 \pm 0$ | $0 \pm 0$ | $\mathbf{0.1} \pm 0.2$ |
| task3-v0 | $1 \pm 1$ | $7 \pm 3$ | $8 \pm 4$ | $3 \pm 1$ | $1 \pm 1$ | $2 \pm 3$ | $1 \pm 1$ | $10 \pm 2$ | $48 \pm 10$ | $11 \pm 7$ | $14.6 \pm 16.2$ |
| task4-v0 | $1 \pm 0$ | $1 \pm 0$ | $1 \pm 1$ | $0 \pm 0$ | $0 \pm 0$ | $0 \pm 1$ | $0 \pm 0$ | $0 \pm 0$ | $1 \pm 1$ | $2 \pm 3$ | $\mathbf{3.7} \pm 4.1$ |
| task5-v0 | $0 \pm 1$ | $1 \pm 1$ | $2 \pm 2$ | $0 \pm 0$ | $0 \pm 0$ | $0 \pm 0$ | $0 \pm 0$ | $1 \pm 1$ | $0 \pm 0$ | $1 \pm 3$ | $13.1 \pm 15.5$ |
| *antsoccer-arena-navigate-singletask* | | | | | | | | | | | |
| task1-v0 | $2 \pm 1$ | $14 \pm 5$ | $0 \pm 0$ | $44 \pm 12$ | $2 \pm 1$ | $1 \pm 3$ | $22 \pm 2$ | $17 \pm 3$ | $61 \pm 25$ | $\mathbf{77} \pm 4$ | $77.0 \pm 1.7$ |
| task2-v0 | $2 \pm 2$ | $17 \pm 7$ | $0 \pm 1$ | $15 \pm 12$ | $3 \pm 1$ | $0 \pm 0$ | $8 \pm 1$ | $8 \pm 2$ | $75 \pm 3$ | $88 \pm 3$ | $\mathbf{91.2} \pm 2.2$ |
| task3-v0 | $0 \pm 0$ | $6 \pm 4$ | $0 \pm 0$ | $0 \pm 0$ | $0 \pm 0$ | $8 \pm 19$ | $11 \pm 5$ | $16 \pm 3$ | $14 \pm 22$ | $\mathbf{61} \pm 6$ | $51.9 \pm 4.9$ |
| task4-v0 | $1 \pm 0$ | $3 \pm 2$ | $0 \pm 0$ | $0 \pm 1$ | $0 \pm 0$ | $0 \pm 0$ | $12 \pm 3$ | $24 \pm 4$ | $16 \pm 9$ | $\mathbf{39} \pm 6$ | $40.2 \pm 4.2$ |
| task5-v0 | $0 \pm 0$ | $2 \pm 2$ | $0 \pm 0$ | $0 \pm 0$ | $0 \pm 0$ | $0 \pm 0$ | $9 \pm 2$ | $15 \pm 4$ | $0 \pm 1$ | $\mathbf{36} \pm 9$ | $29.1 \pm 8.9$ |
| *cube-single-play-singletask* | | | | | | | | | | | |
| task1-v0 | $10 \pm 5$ | $88 \pm 3$ | $89 \pm 5$ | $\mathbf{95} \pm 2$ | $89 \pm 7$ | $77 \pm 28$ | $81 \pm 9$ | $73 \pm 33$ | $79 \pm 4$ | $97 \pm 2$ | $\mathbf{99.1} \pm 0.4$ |
| task2-v0 | $3 \pm 1$ | $85 \pm 8$ | $92 \pm 4$ | $\mathbf{96} \pm 2$ | $82 \pm 16$ | $80 \pm 30$ | $81 \pm 9$ | $83 \pm 13$ | $73 \pm 3$ | $97 \pm 2$ | $\mathbf{99.4} \pm 0.7$ |
| task3-v0 | $9 \pm 5$ | $91 \pm 5$ | $93 \pm 3$ | $\mathbf{99} \pm 1$ | $96 \pm 2$ | $98 \pm 1$ | $87 \pm 4$ | $82 \pm 12$ | $88 \pm 4$ | $98 \pm 2$ | $\mathbf{99.4} \pm 0.5$ |
| task4-v0 | $2 \pm 1$ | $73 \pm 6$ | $92 \pm 3$ | $93 \pm 4$ | $70 \pm 18$ | $91 \pm 2$ | $79 \pm 6$ | $79 \pm 20$ | $79 \pm 6$ | $94 \pm 3$ | $\mathbf{99.1} \pm 0.7$ |
| task5-v0 | $3 \pm 3$ | $78 \pm 9$ | $87 \pm 8$ | $90 \pm 6$ | $61 \pm 12$ | $80 \pm 20$ | $78 \pm 10$ | $76 \pm 33$ | $77 \pm 7$ | $93 \pm 3$ | $\mathbf{97.0} \pm 1.6$ |
| *cube-double-play-singletask* | | | | | | | | | | | |
| task1-v0 | $8 \pm 3$ | $27 \pm 5$ | $45 \pm 6$ | $39 \pm 19$ | $7 \pm 6$ | $21 \pm 8$ | $21 \pm 7$ | $47 \pm 11$ | $35 \pm 9$ | $61 \pm 9$ | $\mathbf{76.1} \pm 4.6$ |
| task2-v0 | $0 \pm 0$ | $1 \pm 1$ | $7 \pm 3$ | $16 \pm 10$ | $0 \pm 0$ | $2 \pm 2$ | $2 \pm 1$ | $22 \pm 12$ | $9 \pm 5$ | $36 \pm 6$ | $\mathbf{53.3} \pm 8.9$ |
| task3-v0 | $0 \pm 0$ | $0 \pm 0$ | $4 \pm 1$ | $17 \pm 8$ | $0 \pm 1$ | $3 \pm 1$ | $1 \pm 1$ | $4 \pm 2$ | $8 \pm 5$ | $22 \pm 5$ | $\mathbf{43.3} \pm 8.9$ |
| task4-v0 | $0 \pm 0$ | $0 \pm 0$ | $1 \pm 1$ | $0 \pm 1$ | $0 \pm 0$ | $0 \pm 1$ | $0 \pm 0$ | $0 \pm 1$ | $1 \pm 1$ | $5 \pm 2$ | $\mathbf{7.1} \pm 3.1$ |
| task5-v0 | $0 \pm 0$ | $4 \pm 3$ | $4 \pm 2$ | $1 \pm 1$ | $0 \pm 0$ | $3 \pm 2$ | $2 \pm 1$ | $2 \pm 2$ | $17 \pm 6$ | $19 \pm 10$ | $\mathbf{56.3} \pm 11.3$ |
| *scene-play-singletask* | | | | | | | | | | | |
| task1-v0 | $19 \pm 6$ | $94 \pm 3$ | $\mathbf{95} \pm 2$ | $100 \pm 0$ | $94 \pm 4$ | $100 \pm 1$ | $87 \pm 8$ | $96 \pm 8$ | $98 \pm 3$ | $100 \pm 0$ | $\mathbf{99.8} \pm 0.4$ |
| task2-v0 | $1 \pm 1$ | $12 \pm 3$ | $50 \pm 13$ | $33 \pm 14$ | $2 \pm 2$ | $50 \pm 40$ | $18 \pm 8$ | $46 \pm 10$ | $0 \pm 0$ | $76 \pm 9$ | $\mathbf{89.0} \pm 4.1$ |
| task3-v0 | $1 \pm 1$ | $32 \pm 7$ | $55 \pm 16$ | $\mathbf{94} \pm 4$ | $4 \pm 4$ | $49 \pm 16$ | $38 \pm 9$ | $78 \pm 14$ | $54 \pm 19$ | $98 \pm 1$ | $78.0 \pm 13.2$ |
| task4-v0 | $2 \pm 2$ | $0 \pm 1$ | $3 \pm 3$ | $4 \pm 3$ | $0 \pm 0$ | $0 \pm 0$ | $\mathbf{6} \pm 1$ | $4 \pm 4$ | $0 \pm 0$ | $5 \pm 1$ | $0.6 \pm 0.6$ |
| task5-v0 | $0 \pm 0$ | $0 \pm 0$ | $0 \pm 0$ | $0 \pm 0$ | $0 \pm 0$ | $0 \pm 0$ | $0 \pm 0$ | $0 \pm 0$ | $0 \pm 0$ | $0 \pm 0$ | $0.0 \pm 0.0$ |
| *puzzle-3x3-play-singletask* | | | | | | | | | | | |
| task1-v0 | $5 \pm 2$ | $33 \pm 6$ | $\mathbf{97} \pm 4$ | $52 \pm 12$ | $89 \pm 5$ | $97 \pm 2$ | $25 \pm 9$ | $63 \pm 19$ | $94 \pm 3$ | $90 \pm 4$ | $\mathbf{94.8} \pm 4.4$ |
| task2-v0 | $1 \pm 1$ | $4 \pm 3$ | $1 \pm 1$ | $0 \pm 1$ | $0 \pm 1$ | $0 \pm 0$ | $4 \pm 2$ | $2 \pm 2$ | $1 \pm 2$ | $\mathbf{16} \pm 5$ | $0.3 \pm 0.3$ |
| task3-v0 | $1 \pm 1$ | $3 \pm 2$ | $3 \pm 1$ | $0 \pm 0$ | $0 \pm 0$ | $0 \pm 0$ | $1 \pm 0$ | $1 \pm 1$ | $0 \pm 0$ | $\mathbf{10} \pm 3$ | $0.9 \pm 0.6$ |
| task4-v0 | $1 \pm 1$ | $2 \pm 1$ | $2 \pm 1$ | $0 \pm 0$ | $0 \pm 0$ | $0 \pm 0$ | $1 \pm 1$ | $2 \pm 2$ | $0 \pm 0$ | $\mathbf{16} \pm 5$ | $5.4 \pm 2.1$ |
| task5-v0 | $1 \pm 0$ | $3 \pm 2$ | $5 \pm 3$ | $0 \pm 0$ | $0 \pm 0$ | $0 \pm 0$ | $1 \pm 1$ | $2 \pm 2$ | $0 \pm 0$ | $\mathbf{16} \pm 3$ | $14.1 \pm 8.4$ |
| *puzzle-4x4-play-singletask* | | | | | | | | | | | |
| task1-v0 | $1 \pm 1$ | $12 \pm 2$ | $26 \pm 4$ | $48 \pm 5$ | $24 \pm 9$ | $44 \pm 10$ | $1 \pm 2$ | $32 \pm 9$ | $49 \pm 9$ | $34 \pm 8$ | $\mathbf{50.0} \pm 8.1$ |
| task2-v0 | $0 \pm 0$ | $7 \pm 4$ | $12 \pm 4$ | $14 \pm 5$ | $0 \pm 1$ | $0 \pm 0$ | $0 \pm 1$ | $5 \pm 3$ | $4 \pm 4$ | $\mathbf{16} \pm 5$ | $9.9 \pm 2.5$ |
| task3-v0 | $0 \pm 0$ | $9 \pm 3$ | $15 \pm 3$ | $34 \pm 5$ | $21 \pm 10$ | $29 \pm 12$ | $1 \pm 1$ | $20 \pm 10$ | $\mathbf{50} \pm 14$ | $18 \pm 5$ | $46.2 \pm 3.8$ |
| task4-v0 | $0 \pm 0$ | $5 \pm 2$ | $10 \pm 3$ | $\mathbf{26} \pm 6$ | $7 \pm 4$ | $1 \pm 1$ | $0 \pm 0$ | $5 \pm 1$ | $21 \pm 11$ | $11 \pm 3$ | $17.2 \pm 2.5$ |
| task5-v0 | $0 \pm 0$ | $4 \pm 1$ | $7 \pm 3$ | $\mathbf{24} \pm 11$ | $1 \pm 1$ | $0 \pm 0$ | $0 \pm 1$ | $4 \pm 3$ | $2 \pm 2$ | $7 \pm 3$ | $7.3 \pm 3.6$ |
| **Average (50 tasks)** | 2.8 | 23.4 | 30.9 | 22.8 | 14.2 | 24.7 | 16.1 | 28.0 | 30.0 | 43.5 | **51.8** |

Table 11: **Offline RL full results on D4RL.** For each task, models were trained with 8 random seeds and evaluated at the end of training. Reported values are the average normalized scores over the final 100 evaluation episodes, with $\pm$ denoting the standard deviation across seeds. *Italics* indicate scores from prior work (Fu et al., 2020; Fujimoto & Gu, 2021; Tarasov et al., 2023; Park et al., 2025), and bold denotes values within 95% of the best performance. GFP actor $\pi_\theta$ is our primary policy, while GFP VaBC $\pi_\omega$ is reported as a byproduct of training.

| Task | Offline RL algorithms | | | | | | | |
|------|------|------|------|------|------|------|------|------|
| | BC | CQL | IQL | TD3 + BC | ReBRAC | FQL | **GFP** actor $\pi_\omega$ | **GFP** VaBC $\pi_\theta$ |
| D4RL antmaze-umaze | *55* | *74.0* | *87.5* | *78.6* | $\boldsymbol{97.8} \pm 1.0$ | $\boldsymbol{96} \pm 2$ | $96.8 \pm 1.9$ | $94.9 \pm 2.0$ |
| D4RL antmaze-umaze-diverse | *47* | *84.0* | *62.2* | *71.4* | $\boldsymbol{88.3} \pm 13.0$ | $\boldsymbol{89} \pm 5$ | $91.9 \pm 2.7$ | $90.1 \pm 3.8$ |
| D4RL antmaze-medium-play | *0* | *61.2* | *71.2* | *3.0* | $\boldsymbol{84.0} \pm 4.2$ | $78 \pm 7$ | $81.9 \pm 5.2$ | $57.4 \pm 9.1$ |
| D4RL antmaze-medium-diverse | *1* | *53.7* | *70.0* | *10.6* | $\boldsymbol{76.3} \pm 13.5$ | $71 \pm 13$ | $61.6 \pm 20.9$ | $45.6 \pm 9.5$ |
| D4RL antmaze-large-play | *0* | *15.8* | *39.6* | *0.0* | $60.4 \pm 26.1$ | $\boldsymbol{84} \pm 7$ | $82.6 \pm 5.4$ | $62.6 \pm 8.8$ |
| D4RL antmaze-large-diverse | *0* | *14.9* | *47.5* | *0.2* | $54.4 \pm 25.1$ | $\boldsymbol{83} \pm 4$ | $84.1 \pm 5.4$ | $70.6 \pm 4.7$ |
| D4RL pen-human-v1 | *71* | *37.5* | *71.5* | *81.8* | $\boldsymbol{103.5}$ | *53* | $64.6 \pm 5.4$ | $67.4 \pm 6.9$ |
| D4RL pen-cloned-v1 | *52* | *39.2* | *37.3* | *61.4* | $\boldsymbol{91.8}$ | *74* | $77.1 \pm 10.4$ | $70.5 \pm 4.2$ |
| D4RL pen-expert-v1 | *110* | *107.0* | *133.6* | *146* | $\boldsymbol{154.1}$ | *142* | $140.4 \pm 4.7$ | $123.2 \pm 5.4$ |
| D4RL door-human-v1 | *2* | $\boldsymbol{9.9}$ | *4.3* | *-0.1* | *0.0* | *0.0* | $0.3 \pm 0.3$ | $4.1 \pm 2.4$ |
| D4RL door-cloned-v1 | *-0* | *0.4* | $\boldsymbol{1.6}$ | *0.1* | *1.1* | *2* | $\boldsymbol{1.6} \pm 1.9$ | $0.6 \pm 0.6$ |
| D4RL door-expert-v1 | *105* | *101.5* | $\boldsymbol{105.3}$ | *84.6* | $\boldsymbol{104.6}$ | *104* | $\boldsymbol{104.1} \pm 0.6$ | $\boldsymbol{103.1} \pm 0.9$ |
| D4RL hammer-human-v1 | *3* | $\boldsymbol{4.4}$ | *1.4* | *0.4* | *0.2* | *1* | $\boldsymbol{4.4} \pm 4.9$ | $2.5 \pm 1.1$ |
| D4RL hammer-cloned-v1 | *1* | *2.1* | *2.1* | *0.8* | *6.7* | $\boldsymbol{11}$ | $\boldsymbol{12.4} \pm 5.4$ | $2.5 \pm 0.9$ |
| D4RL hammer-expert-v1 | *127* | *86.7* | $\boldsymbol{129.6}$ | *117.0* | $\boldsymbol{133.8}$ | *125* | $123.6 \pm 2.0$ | $116.6 \pm 4.1$ |
| D4RL relocate-human-v1 | *0* | *0.20* | *0.1* | *-0.2* | *0.0* | *0* | $0.5 \pm 0.3$ | $0.0 \pm 0.0$ |
| D4RL relocate-cloned-v1 | *-0* | *-0.1* | *-0.2* | *-0.1* | $\boldsymbol{1.9}$ | *-0* | $1.6 \pm 0.7$ | $0.1 \pm 0.1$ |
| D4RL relocate-expert-v1 | $\boldsymbol{108}$ | *95.0* | $\boldsymbol{106.5}$ | $\boldsymbol{107.3}$ | $\boldsymbol{106.6}$ | $\boldsymbol{107}$ | $\boldsymbol{103.2} \pm 3.7$ | $\boldsymbol{104.0} \pm 3.1$ |

Table 12: **Offline RL full results on Minari.** For each task, models were trained using 8 different random seeds, and evaluation was performed at the end of training. The reported values represent the average normalized score, computed over the final 100 evaluation episodes and averaged across the 8 seeds. GFP actor $\pi_\theta$ represents our primary policy, while GFP VaBC $\pi_\omega$ is reported as a byproduct of our training procedure.

| Task | Offline RL algorithms | | |
|------|------|------|------|
| | FQL | **GFP** actor $\pi_\theta$ | **GFP** VaBC $\pi_\omega$ |
| Minari pen-human-v2 | $11.5 \pm 4.9$ | $50.1 \pm 6.3$ | $\boldsymbol{54.4} \pm 6.7$ |
| Minari pen-cloned-v2 | $41.8 \pm 3.7$ | $\boldsymbol{60.4} \pm 4.2$ | $54.0 \pm 5.5$ |
| Minari pen-expert-v2 | $92.7 \pm 6.2$ | $\boldsymbol{115.9} \pm 4.5$ | $108.7 \pm 2.8$ |
| Minari door-human-v2 | $1.1 \pm 0.5$ | $0.4 \pm 0.1$ | $\boldsymbol{2.7} \pm 1.9$ |
| Minari door-cloned-v2 | $\boldsymbol{0.4} \pm 0.2$ | $0.3 \pm 0.3$ | $0.1 \pm 0.1$ |
| Minari door-expert-v2 | $\boldsymbol{102.0} \pm 0.9$ | $94.1 \pm 20.9$ | $99.0 \pm 10.3$ |
| Minari hammer-human-v2 | $1.0 \pm 0.6$ | $2.7 \pm 0.8$ | $\boldsymbol{3.1} \pm 0.8$ |
| Minari hammer-cloned-v2 | $1.0 \pm 0.6$ | $\boldsymbol{22.9} \pm 19.5$ | $5.9 \pm 4.1$ |
| Minari hammer-expert-v2 | $121.2 \pm 4.2$ | $\boldsymbol{130.2} \pm 5.4$ | $119.0 \pm 6.7$ |
| Minari relocate-human-v2 | $-0.0 \pm 0.0$ | $\boldsymbol{0.1} \pm 0.2$ | $-0.0 \pm 0.1$ |
| Minari relocate-cloned-v2 | $0.0 \pm 0.0$ | $\boldsymbol{0.3} \pm 0.2$ | $0.0 \pm 0.0$ |
| Minari relocate-expert-v2 | $\boldsymbol{103.7} \pm 1.2$ | $102.1 \pm 5.3$ | $\boldsymbol{105.9} \pm 1.2$ |
| Minari halfcheetah-simple-v0 | $59.2 \pm 0.3$ | $\boldsymbol{72.5} \pm 0.5$ | $64.4 \pm 0.4$ |
| Minari halfcheetah-medium-v0 | $100.2 \pm 6.7$ | $\boldsymbol{121.3} \pm 5.8$ | $108.6 \pm 5.3$ |
| Minari halfcheetah-expert-v0 | $\boldsymbol{134.1} \pm 2.3$ | $133.4 \pm 1.3$ | $\boldsymbol{136.4} \pm 0.8$ |
| Minari hopper-simple-v0 | $57.2 \pm 5.9$ | $\boldsymbol{91.6} \pm 4.3$ | $\boldsymbol{87.4} \pm 6.6$ |
| Minari hopper-medium-v0 | $\boldsymbol{81.9} \pm 23.9$ | $\boldsymbol{79.6} \pm 14.5$ | $\boldsymbol{78.2} \pm 24.2$ |
| Minari hopper-expert-v0 | $99.6 \pm 10.9$ | $\boldsymbol{103.9} \pm 10.6$ | $\boldsymbol{108.8} \pm 13.1$ |
| Minari walker2d-simple-v0 | $\boldsymbol{89.4} \pm 0.7$ | $\boldsymbol{90.0} \pm 0.9$ | $\boldsymbol{89.7} \pm 0.9$ |
| Minari walker2d-medium-v0 | $127.6 \pm 3.0$ | $\boldsymbol{133.7} \pm 1.1$ | $126.1 \pm 3.5$ |
| Minari walker2d-expert-v0 | $148.0 \pm 1.8$ | $149.9 \pm 2.0$ | $\boldsymbol{150.8} \pm 0.4$ |

Table 13: **Task-specific hyperparameters for offline RL on OGBench.**

| Task Category | Offline RL algorithms | | | | |
|---|---|---|---|---|---|
| | IQL $\alpha$ | ReBRAC $(\alpha_1,\ \alpha_2)$ | FQL $\alpha$ | GFP (ours) $(\alpha,\ \eta)$ | GFP-AWR Sec. B.3 $(\alpha,\ \eta)$ |
| antmaze-large-navigate-singletask-task{1,2,3,4,5}-v0 | 1e+1 | (1e−2, 1e−2) | 1e+1 | (3e−1, 1e−4) | – |
| antmaze-large-stitch-singletask-task{1,2,3,4,5}-v0 | 1e+1 | (1e−2, 1e−2) | 3e+0 | (3e−2, 1e−6) | (1e−1, 3e−1) |
| antmaze-large-explore-singletask-task{1,2,3,4,5}-v0 | 1e+0 | (1e−3, 1e−1) | 1e+0 | (1e−2, 1e−6) | – |
| antmaze-large-giant-singletask-task{1,2,3,4,5}-v0 | – | (1e−2, 1e−2) | 3e+1 | (1e−1, 1e−1) | (1e−1, 3e−1) |
| humanoidmaze-medium-navigate-singletask-task{1,2,3,4,5}-v0 | – | (1e−2, 1e−2) | 3e+1 | (3e−1, 1e−3) | (3e−1, 3e−1) |
| humanoidmaze-medium-stitch-singletask-task{1,2,3,4,5}-v0 | 1e+1 | (1e−2, 1e−2) | 1e+2 | (3e−1, 1e−3) | (3e−1, 1e−1) |
| humanoidmaze-large-navigate-singletask-task{1,2,3,4,5}-v0 | – | (1e−2, 0e+0) | 1e+2 | (3e−1, 1e−4) | (3e−1, 1e−1) |
| antsoccer-arena-navigate-singletask-task{1,2,3,4,5}-v0 | – | (1e−2, 0e+0) | – | (1e−1, 1e−2) | – |
| antsoccer-arena-stitch-singletask-task{1,2,3,4,5}-v0 | 1e+0 | (1e−2, 1e−3) | 1e+1 | (1e−1, 1e−2) | (1e−1, 1e−1) |
| cube-single-play-singletask-task{1,2,3,4,5}-v0 | 1 | – | – | (1e+1, 1e−1) | – |
| cube-single-noisy-singletask-task{1,2,3,4,5}-v0 | 3e+0 | (1e−1, 1e−1) | 3e+1 | (1e+1, 1e−3) | – |
| cube-double-play-singletask-task{1,2,3,4,5}-v0 | – | (1e−1, 3e−1) | – | (1e+0, 1e−2) | (1e+0, 1e−1) |
| cube-double-noisy-singletask-task{1,2,3,4,5}-v0 | 3e−1 | (1e−2, 1e−2) | 1e+1 | (1e−1, 1e−4) | (1e−1, 1e−1) |
| cube-triple-play-singletask-task{1,2,3,4,5}-v0 | 1e+0 | (1e−1, 1e−3) | 3e+2 | (1e+0, 1e−5) | (1e+0, 3e+0) |
| cube-triple-noisy-singletask-task{1,2,3,4,5}-v0 | 3e+0 | (1e−2, 0e+0) | 1e+1 | (1e−1, 1e−5) | (3e−2, 1e+0) |
| scene-play-singletask-task{1,2,3,4,5}-v0 | – | (1e−1, 1e−3) | – | (1e+1, 1e−3) | – |
| scene-noisy-singletask-task{1,2,3,4,5}-v0 | 1e+1 | (3e−3, 0e+0) | 3e+1 | (1e+0, 1e−4) | – |
| puzzle-3×3-play-singletask-task{1,2,3,4,5}-v0 | – | – | – | (3e+0, 1e−3) | (3e+0, 1e+0) |
| puzzle-4×4-play-singletask-task{1,2,3,4,5}-v0 | – | (1e−1, 0e+0) | – | (3e+0, 1e−5) | (1e+0, 3e−1) |
| puzzle-4×4-noisy-singletask-task{1,2,3,4,5}-v0 | 1e+0 | (3e−2, 1e−2) | 3e+2 | (3e+0, 1e−3) | (3e−1, 1e−1) |
| visual-cube-single-play-singletask-task1-v0 | – | – | – | (1e+1, 1e−1) | – |
| visual-cube-double-play-singletask-task1-v0 | – | – | – | (3e−1, 1e−2) | – |
| visual-scene-play-singletask-task1-v0 | – | – | – | (1e+1, 1e−3) | – |
| visual-puzzle-3x3-play-singletask-task1-v0 | – | – | – | (3e+0, 1e−2) | – |
| visual-puzzle-4x4-play-singletask-task1-v0 | – | – | – | (1e+0, 1e−4) | – |

Table 14: **Task-specific hyperparameters for offline RL on D4RL and Minari.**

| Task | GFP (ours) $(\alpha,\ \eta)$ |
|---|---|
| D4RL antmaze-umaze | (1e−1, 1e−3) |
| D4RL antmaze-umaze-diverse | (1e−1, 1e−3) |
| D4RL antmaze-medium-play | (3e−2, 1e−3) |
| D4RL antmaze-medium-diverse | (3e−2, 1e−3) |
| D4RL antmaze-large-play | (3e−2, 1e−5) |
| D4RL antmaze-large-diverse | (3e−2, 1e−5) |
| D4RL pen-human-v1 | (3e+0, 1e−4) |
| D4RL pen-cloned-v1 | (3e+0, 1e−5) |
| D4RL pen-expert-v1 | (1e+0, 1e−3) |
| D4RL door-human-v1 | (1e+1, 1e−2) |
| D4RL door-cloned-v1 | (1e+1, 1e−2) |
| D4RL door-expert-v1 | (1e+1, 1e−2) |
| D4RL hammer-human-v1 | (1e+1, 1e−5) |
| D4RL hammer-cloned-v1 | (1e+1, 1e−5) |
| D4RL hammer-expert-v1 | (1e+1, 1e−2) |
| D4RL relocate-human-v1 | (1e+1, 1e−4) |
| D4RL relocate-cloned-v1 | (1e+1, 1e−4) |
| D4RL relocate-expert-v1 | (1e+1, 1e−4) |

(a) D4RL

| Task | FQL $\alpha$ | GFP (ours) $(\alpha,\ \eta)$ |
|---|---|---|
| Minari pen-human-v2 | 1e+4 | (3e+0, 1e−6) |
| Minari pen-cloned-v2 | 3e+3 | (1e+0, 1e−4) |
| Minari pen-expert-v2 | 1e+3 | (3e−1, 1e−6) |
| Minari door-human-v2 | 3e+4 | (1e−2, 1e−2) |
| Minari door-cloned-v2 | 3e+4 | (3e−2, 1e−6) |
| Minari door-expert-v2 | 3e+4 | (3e+0, 1e−3) |
| Minari hammer-human-v2 | 3e+4 | (1e+0, 1e−2) |
| Minari hammer-cloned-v2 | 3e+4 | (3e+0, 1e−5) |
| Minari hammer-expert-v2 | 1e+4 | (1e+0, 1e−5) |
| Minari relocate-human-v2 | 3e+3 | (3e+0, 1e−5) |
| Minari relocate-cloned-v2 | 3e+4 | (3e+0, 1e−4) |
| Minari relocate-expert-v2 | 3e+4 | (3e+0, 1e−3) |
| Minari halfcheetah-simple-v0 | 1e+2 | (3e−2, 1e−6) |
| Minari halfcheetah-medium-v0 | 1e+2 | (1e−1, 1e−3) |
| Minari halfcheetah-expert-v0 | 1e+3 | (3e+0, 1e−2) |
| Minari hopper-simple-v0 | 3e+2 | (3e+0, 1e−3) |
| Minari hopper-medium-v0 | 3e+2 | (3e+0, 1e−3) |
| Minari hopper-expert-v0 | 1e+3 | (3e+0, 1e−2) |
| Minari walker2d-simple-v0 | 1e+3 | (3e+0, 1e−1) |
| Minari walker2d-medium-v0 | 3e+2 | (3e−1, 1e−2) |
| Minari walker2d-expert-v0 | 3e+2 | (3e+0, 1e−3) |

(b) Minari

