# OpenReview forum: "Guided Flow Policy: Learning from High-Value Actions in Offline Reinforcement Learning"
_ICLR.cc/2026/Conference — ICLR 2026 Poster_

### Official Review · Reviewer_Zk33 · 2025-10-24

**Soundness:** 3
**Presentation:** 3
**Contribution:** 2
**Rating:** 6
**Confidence:** 4

**Summary:**

This paper proposes Guided Flow Policy (GFP), an offline RL algorithm that trains a flow policy. The method is built on top of the behavioral regularized actor-critic framework, and more specifically, flow Q-learning (FQL). The main differences between GFP and FQL are that (1) GFP uses a weighted BC flow policy instead of a vanilla BC policy, and (2) it uses a more conservative Q target by mixing the behavioral action and the policy action. The authors evaluate their method on a large number (129) of tasks across OGBench, Minari, and D4RL, showing that GFP outperforms FQL and other strong baselines.

**Strengths:**

* The paper is well-written and easy to understand.
* The empirical evaluation is notably thorough. Across more than 100 tasks, GFP consistently achieves better performance than other baselines (IQL, FQL, and ReBRAC). The authors also mainly use 8 seeds for each task.
* The paper provides several ablation studies, including analyses of the effects of conservative Q targets, weighted behavioral cloning, etc. They demonstrate the importance of these techniques.

**Weaknesses:**

I don't see major weaknesses in this work. Two minor ones are as follows:
* One weakness is its relatively limited novelty and contribution. Essentially, the paper can be summarized as AWR + FQL (or broadly AWR + BRAC), which is a rather straightforward extension of existing methods. While I do appreciate the simplicity of this combination, given how straightforward the method is, the paper could further be strengthened by either having more large-scale experiments (e.g., more challenging/long-horizon tasks, more realistic benchmarks (or even real robot experiments), pixel-based tasks, etc.) and/or providing additional insights beyond simply applying existing techniques to the flow RL setting.
* Another weakness is that this method requires an additional parameter ($\eta$) that needs to be tuned per task, compared to FQL. Generally, introducing more hyperparameters would naturally only make the method better, so the real question is how large the gap is compared to the closest baseline (FQL) with this extra degree of freedom. From Figure 4 and Table 2, the difference doesn't seem unreasonably small, but it would have been even more convincing if the proposed method had achieved an even larger performance boost.

**Questions:**

Have you tried vanilla AWR for learning a weighted BC policy? Why do we want to have the additional normalization term in the denominator (compared to AWR) in Eq. (10)?

---

> ### Author Response · Authors · 2025-11-24
> **Response to reviewer Zk33's initial review**
>
> We thank the reviewer for their positive assessment of our work. We are particularly encouraged that you recognized the thoroughness of our empirical evaluation and found the paper well-written.
>
> ### 1. Response to Weakness 1: Novelty and Contribution
>
> > **Reviewer Point:** The method can be summarized as AWR + FQL... a straightforward extension... could be strengthened by having more large-scale/challenging experiments.
>
> We acknowledge the reviewer's observation that our method combines existing mechanisms (AWR-style weighting and Flow Matching), and that the method is conceptually simple (which we also see as a strength). However, we respectfully argue that finding the _right_ combination of existing mechanisms that works robustly across diverse domains is a significant challenge. In apology for a lack of positioning with respect to related works in the original submission, we revised the paper, framing the core contribution as effectively combining two predominant policy extraction methods (weighted BC and behavior regularized policy gradients) with bidirectional interactions. Modifications are listed in the general answer to all reviewers.
>
> Specifically to address the reviewer's suggestion to expand our evaluations we ran new experiments:
> - We first added a comparison to 10 offline-RL prior works, on the 50 tasks previously evaluated by FQL.
> - We expanded our extensive set of experiments to 144 tasks, including 5 pixel-based tasks, in particular, on the challenging cube-triple-noisy environment GFP achieves an average score of 24.5 compared to 4.8 for IQL, 5.2 for ReBRAC and 3.5 for FQL (Tab 2 and Tab 9).
>
>
> ### 2. Response to Weakness 2: Hyperparameter Sensitivity ($\eta$)
>
> > **Reviewer Point:** The method requires an additional parameter ($\eta$) ... the real question is how large the gap is compared to FQL with this extra degree of freedom.
>
> We agree that introducing hyperparameters requires careful justification. However, the parameter $\eta$ provides a necessary control over the conservatism of the behaviour cloning, allowing the algorithm to effectively filter out noisy actions in suboptimal datasets. To convince the reviewer, we want to highlight GFP's performance on challenging tasks. On the cube-triple-noisy environment previously mentioned, on task 1, GFP achieves a score of 90.7 compared to 17.5 for FQL and 25.3 for ReBRAC, see Tab 9.
>
> **Note:** We additionally did new experiments to measure the sensitivity of GFP to both $\alpha$, the BC coefficient, and $\eta$ the temperature. We added a new subsection in the appendix, Sec B.5 and Fig 7, with a reference to it Sec 4.2. For 12 tasks, we took the task-specific hyperparameters $(\alpha,\eta)$ and we tried all 8 variations by increasing or decreasing either or both alpha and eta. This study helps clarify the sensitivity and interdependencies of these hyperparameters. We find that the sensitivity on $\alpha$ (similar to FQL and ReBrac) is much higher than the one on $\eta$ (newly introduced). It therefore does not complicate the tuning of the method much.
>
>
> ### 3. Response to Specific Questions
>
> > Q1: Have you tried vanilla AWR for learning a weighted BC policy?
>
> GFP weighting was inspired by a soft-max approach but the reviewer rightly pointed out a connection with AWR-style weighting. We added a paragraph at the end of Sec 3 to clarify this relation with AWR and we conducted new experiments on 65 tasks to compare GFP soft-max style weighting to AWR-style (in the appendix, Sec B.2). We warmly thank the reviewer for this suggestion, as we believe it helped us improve the paper.
>
>
> > Q2: Why do we want to have the additional normalization term in the denominator (compared to AWR) in Eq. (10)?
>
> Standard AWR implementations rely on heuristic clipping of the importance weights (exponentials) to maintain numerical stability and prevent exploding gradients.
>
> As stated in [1]: " The weights $\omega$ used to update the policy can occasionally assume excessively large values, which can cause gradients to explode. We therefore apply weight clipping with a threshold $\omega_{\text{max}}$ to mitigate issues due to exploding weights."
>
> Our normalization term follows a similar objective, to restrain the weight values to stay between 0 and 1.
>
> [1] Peng, X.B., Kumar, A., Zhang, G., & Levine, S. (2019). Advantage-Weighted Regression: Simple and Scalable Off-Policy Reinforcement Learning. ArXiv, abs/1910.00177.

---

> > ### Comment · Reviewer_Zk33 · 2025-11-26
> >
> > Thanks for the response. I appreciate the new results and clarifications. I don't have any outstanding concerns, and would like to maintain my initial evaluation of a slightly positive score.

---

### Official Review · Reviewer_AGZW · 2025-10-26

**Soundness:** 3
**Presentation:** 3
**Contribution:** 2
**Rating:** 2
**Confidence:** 4

**Summary:**

The paper proposes guided flow policy (GFP), an offline RL method that leverages flow-matching to train expressive policies that can tackle tasks with suboptimal datasets. The proposed method trains the base flow-matching policy  $\pi_\omega$ via weighted behavior cloning to match $\pi(a\mid s) \propto_a \pi_\beta \exp(\tau Q(s, a))$ (where $\pi_\beta$ is the behavior distribution described by the dataset). Then, it applies a recently proposed technique, FQL, on top of $\pi_\omega$ where it trains a 1-step distilled policy to stay close to the base flow policy $\pi_\omega$ while maximizing its value under the critic. In practice, the authors adopt a slightly different/more conservative Q-target backup by using a mixture of actions from both the more aggressive 1-step distilled policy and the more conservative multi-step base flow policy. Across multiple benchmarks (OGBench, D4RL, Minari), the proposed method outperforms FQL and Gaussian policy offline RL baselines (ReBRAC, IQL) especially on noisier offline datasets.

**Strengths:**

- The paper considers a comprehensive set of benchmark tasks which make the comparison between the proposed method and the baselines very convincing. I also appreciate that the authors seem to properly tune the hyperparameters for the baselines, which make the comparisons fair.

- The proposed method is presented in a concise and clear manner and the algorithmic design has no technical issues.

**Weaknesses:**

The paper make some factually questionable claims
- Table 1: all prior methods (IQL, TD3+BC, ReBRAC, FQL) are listed under "handles suboptimal data (x)" which implies that these methods cannot handle suboptimal data. This is misleading because most offline RL methods listed and in fact most offline RL methods in general can handle suboptimal data. It would be good to further clarify and be precise about what "handles suboptimal data (x)" actually means here.
- The authors introduce Value-aware behavior cloning (VaBC) as a novelty in the algorithm and claim that "this is the first integration of value-aware behavior cloning", but as how it is currently presented it seems to be me that it is exactly equivalent to advantage weighted regression applied to flow-matching, which has been explored in many recent flow-matching/diffusion papers (e.g., FAWAC baseline in FQL[1], energy-weighted flow matching loss in QIPO [2]).

The comparison to prior work is lacking:
- The paper only compares with a very limited number of baselines (FQL is the only baseline for expressive policies). For the similarity to prior work as mentioned above, I would encourage the authors to consider comparing to some of them such that the reviewers could be more informed about where GFP's performance stands in the literature.
- The related work section discusses very little prior work despite a rich literature in diffusion/flow-matching in reinforcement learning and it is not clear from the section how the proposed method is different from these prior work.

Small typos:
   - Table 4: in the value column, second row: "50,0000" should be "500,000"
   - L248: $Q_{\bar \phi}(s', \mu_\omega(s', z)$ is missing a right bracket.

[1] Park, Seohong, Qiyang Li, and Sergey Levine. "Flow q-learning." arXiv preprint arXiv:2502.02538 (2025).

[2] Zhang, Shiyuan, Weitong Zhang, and Quanquan Gu. "Energy-weighted flow matching for offline reinforcement learning." arXiv preprint arXiv:2503.04975 (2025).

**Questions:**

Related to my concern above -- how is the proposed `value-aware behavior cloning' different from prior methods? (e.g., QIPO and FAWAC)

---

> ### Author Response · Authors · 2025-11-24
> **Response to reviewer AGZW's initial review**
>
> We thank the reviewer for acknowledging the clear design of our method and our effort to provide fair comparisons and strong baselines. We also appreciate the reviewer’s detailed comments on factual clarity, novelty and typos, which helped us improve the paper.
>
> ### 1. Response to Weakness 1: Suboptimal data handling
>
> > **Reviewer Point:**  all prior methods [..] cannot handle suboptimal data .. misleading .. good to further clarify and be precise
>
> We agree that the wording “handles suboptimal data (×)” in Table 1 was misleading and appreciate the reviewer pointing this out. We have revised the row label in Table 1 from “Handles suboptimal data” to the more precise **“Handles suboptimal actions in the regularization term”**. We also replace the cross in the IQL cell with "not applicable" (N/A), since IQL is not a BRAC-style method and does not use the regularization pattern we are discussing here.
>
> Our intention was not to claim that prior methods are unable to handle suboptimal datasets. We intended to highlight a more specific design aspect: whether the regularization term explicitly distinguishes between high and low-value actions. In GFP, the VaBC component is designed to emphasize more promising actions in the regularization, whereas, for example, TD3+BC, ReBRAC, and FQL use regularizers that are value agnostic.
>
>
>
> ### 2. Response to Weakness 2: Novelty of VaBC and relation to AWR-style objectives
> > **Reviewer Point:**  introduce Value-aware behavior cloning as a novelty .. equivalent to advantage weighted regression applied to flow-matching, which has been explored in many recent flow-matching/diffusion papers (FAWAC baseline, QIPO)
>
> We apologize for the confusion regarding how we use weighted behavior cloning. We reworked the paper to position GFP within prior works and highlight GFP core contribution (modifications are listed in the general answer to all reviewers). Our intention was not to claim to be the first to apply weighted behavior cloning or advantage weighting in combination with diffusion or flow-matching models. Our contribution is more specific: we integrate a value-aware/AWR-style behavior cloning policy **as a regularizer within a BRAC framework**, jointly trained with other components (actor and critic), and we analyze this coupled design.
>
> GFP effectively combines the two predominant policy extraction methods in offline-RL: weighted behavior cloning and behavior regularized policy gradients. In particular, we added an explanation at the end of Sec 3/step 3 to clarify the relation with AWR. VaBC is the regularization component of GFP and is trained using a weighting close to advantage weighting, but VaBC is named with respect to its role in GFP structure. The main purpose of VaBC is to add value-awareness in the regularization term, and other weight or extraction method (e.g. rejection sampling) could have been used, as we now discuss in the revised related works Sec 5.
>
> We have also added explicit discussion of related AWR-style weighted flow/diffusion methods (e.g., FAWAC [1], QIPO [2]) in the related work section. Importantly, VaBC differs as it relies on the one-step actor, trained by behavior regularized policy gradients (which has been found to be the most effective policy extraction method [3]), to compute its weights. We also want to clarify that, even though VaBC shares similarities with FAWAC and QIPO as some kind of weighted BC, it differs in (i) its role (as a regularizer inside a BRAC-style actor–critic scheme rather than the final policy), and (ii) its joint, bidirectional coupling with the actor and critic in GFP’s overall training procedure.
>
> **Note**: GFP weighting was inspired by a soft-max approach, but the reviewer rightly pointed out a connection with AWR-style weighting. We added a paragraph at the end of Sec 3 to clarify this relation with AWR, and we conducted new experiments on 65 tasks to compare GFP soft-max style weighting to AWR-style (in the appendix, Sec B.2).
>
> ### 3. Response to Weakness 3: Comparison to prior work
> > **Reviewer Point:**  paper only compares with a very limited number of baselines
>
> We appreciate this feedback. We now include a new comparison to 10 offline-RL prior methods (including FAWAC), on the 50 tasks previously evaluated in the FQL paper [1]. Fig 3a synthetize these results in the new paragraph Sec 4.1 "Comparison against previous works", with full numbers Tab 10. In particular, FAWAC obtains an average score of 16.1 over the 50 tasks, while GFP achieves a score of 51.8 (Tab 10 last row). We thank the reviewer as we believe this additional comparison to numerous methods strengthens the evaluation, helping readers to see where GFP's performance stands in the literature.

---

> > ### Author Response · Authors · 2025-11-24
> > **Response : part 2**
> >
> > ### 4. Response to Weakness 4: Related work
> >
> > > **Reviewer Point:**  section discusses very little prior work despite a rich literature .. not clear from the section how the proposed method is different from these prior work
> >
> > We agree that properly contextualizing GFP within the rich offline RL literature is essential. We apologize for the previously small related work section and the authors thank the reviewer for pointing out this issue. We have significantly expanded the Related Work section to include a broader discussion of advantage-weighted methods (e.g., AWR, AWAC) and recent generative diffusion/flow-matching approaches in RL. Crucially, we have added explicit comparisons to clarify how GFP differs from these families of methods.
> >
> >
> > [1] Park, S., Li, Q., & Levine, S. (2025). Flow Q-Learning. ArXiv, abs/2502.02538.
> >
> > [2] Zhang, S., Zhang, W., & Gu, Q. (2025). Energy-Weighted Flow Matching for Offline Reinforcement Learning. ArXiv, abs/2503.04975.
> >
> > [3] Park, S., Frans, K., Levine, S., & Kumar, A. (2024). Is value learning really the main bottleneck in offline RL?. Advances in Neural Information Processing Systems, 37, 79029-79056.

---

> > > ### Comment · Reviewer_AGZW · 2025-11-25
> > >
> > > Thanks reviewer for the resonse. I appreciate the additional ablation experiments comparing $g^{\mathrm{AWR}}\_\eta(s, a)$ and $g_\eta$ and I think the updated paper is doing a much better job at positioning the GFP in the literature. Overall, I do think that VaBC contains technical novelty compared to prior methods and I am raising my score to 4.
> > >
> > > My main remaining issue is that the paper, at its current form, does not provide sufficient justifications why it is a good idea to do VaBC. While having comprehensive empirical results compared to AWR could make this issue less critical, I am still having doubts/questions regarding the experiments which I will detail below.
> > >
> > > ---
> > >
> > > ## 1. Hyperparameter tuning process
> > >
> > > In Table 4, it is stated that $r + \gamma Q(s’, a’)$ is used by default except on $y^{\mathrm{VaBC}}$. From what I understand, this is a TD-target that only GFP can use and other methods cannot. This could potentially create an unfair comparison (e.g., Table 6 seems to imply that the better of the standard target or the modified target is picked). It would strengthen the paper if the baseline methods also adopt a similar conservative target in the literature (e.g., minimum of two Q functions [1], mean - std * rho of the critic ensemble [2]).
> > >
> > > ---
> > >
> > > ## 2. AWR comparison
> > >
> > > I have noticed some potential issues in your AWR formulation. In your Equation 11, the guidance for AWR is defined as
> > >
> > > $$ g_\eta^{\mathrm{AWR}}(s, a) := \exp\left(\frac{\lambda}{\eta}(Q_\phi(s, a) - Q_\phi(s, \mu_\theta(s, z)))\right). $$
> > >
> > > This is different from the AWR objective in Peng et al. which should be (using the notations in this paper)
> > >
> > > $$ g\_\eta^{\mathrm{AWR}}(s, a) := \exp\left(\frac{\lambda}{\eta}(Q\_\phi(s, a) - V(s)))\right). $$
> > >
> > > where $V(s) = \mathbb{E}\_{(s, a) \sim \mathcal{D}}[Q\_\phi(s, a)]$.
> > >
> > > In your case, $Q_\phi(s, \mu_\theta(s, z))$ is stochastic (because $z$ is sampled from Gaussian) and the optimum of the objective function may not lead to the correct policy (e.g., $\propto \pi_\beta \exp(\frac{\lambda}{\eta} Q_\phi(s, a))$). The same stochasticity problem may also affect your original objective.
> > >
> > > In addition, for your results in Table 7, did you tune GFP-AWR to use either the standard target or $y^{\mathrm{VaBC}}$?
> > >
> > > ---
> > >
> > > ## 3. Characterizations of prior methods
> > >
> > > Table 1 differentiates GFP from prior works by stating that GFP “handles suboptimal actions in the regularization term”. This is a rather vague description and it is not well-motivated why it is crucial to handle suboptimal actions in the regularization term. Furthermore, the table does not contain any advantage-weighted BC-based methods (e.g., FAWAC, QIPO and many more discussed in Weighted behavior cloning paragraph in the related work section).
> > >
> > > In **Positioning GFP within prior work**, the paper says “This bidirectional guidance between the two policies fundamentally distinguishes GFP from other weighted-BC methods: FAWAC Park et al. (2025) relies on the policy itself to compute the state-value for baseline for its weights, and QIPO Zhang et al. (2025) computes weights by sampling multiple actions using the policy itself. In contrast, VaBC relies on the actor to provide baselines for good actions.” — I am a bit confused on what bidirectional guidance means here and I am having a bit of difficulty parsing the description here. Both QIPO and VaBC rely on actor samples to compute the weights for the flow matching loss (e.g., Equation 4.3 QIPO vs. Equation 10 in this paper), so it seems that QIPO also exhibits bidirection guidance.

---

> > > > ### Author Response · Authors · 2025-11-27
> > > > **Response to reviewer AGZW Part 1**
> > > >
> > > > We thank the reviewer for their continued engagement and for raising their score. We are glad that the additional ablations and improved positioning of the paper have clarified the technical novelty of VaBC.
> > > >
> > > > Below, we address your remaining doubts regarding the hyperparameters, the AWR formulation, and the characterization of prior works. **We made a second revision of our paper, and in particular, we added a new figure in the related work section to illustrate an overview of offline-RL methods, page 10, Figure 5**.
> > > >
> > > > ## Remark 1. Hyperparameter Tuning and Target Comparison ($y_{\text{VaBC}}$)
> > > > > Reviewer Point: $y_{\text{VaBC}}$ is a TD-target that only GFP can use... It would strengthen the paper if the baseline methods also adopt a similar conservative target...
> > > >
> > > > We thank the reviewer for highlighting the importance of using appropriate conservative targets also for the baselines. We fully agree that fair comparisons require robust tuning of these targets to ensure strong baselines.
> > > >
> > > > In fact, this is one of the key issues we address in Section 4.3 during our re-evaluation of prior works. All our baseline evaluations employ a double-critic architecture with $(Q_1, Q_2)$, and the important difference lies in how these critics are aggregated. For all re-evaluated methods, we compared the two standard aggregation choices:
> > > >
> > > > $y_{min}^{\text{target}}(s,a) = r(s,a) + \gamma \text{ min }(Q_1(s', \pi_\theta(s')),Q_2(s', \pi_\theta(s')))$
> > > >
> > > > and
> > > > $y_{mean}^{\text{target}}(s,a) = r(s,a) + \frac \gamma 2 (Q_1(s', \pi_\theta(s')),Q_2(s', \pi_\theta(s')))$
> > > >
> > > > Our analysis reveals that the choice of target significantly impacts baseline performance. For example, as shown in Table 3, training ReBRAC on antsoccer-arena-navigate using $y_{\min}^{\text{target}}$ results in a 0% success rate (previously reported), whereas $y_{\text{mean}}^{\text{target}}$ achieves 55.9%. We observe a similar trend on humanoidmaze-medium-navigate (22% vs. 59.2%). This insight helps explain why ReBRAC's performance was underestimated in previous evaluations.
> > > >
> > > > To ensure the fairest possible comparison, we tuned this choice for every environment and every method. And we found this choice to be environment-dependent, but consistent across methods. In Table 4 (row Critic aggregation function), we report which aggregation method was selected to maximize the performance.
> > > >
> > > >
> > > >
> > > > ## Remark 3, Part 1. Characterization of Prior Methods (Table 1)
> > > > > Reviewer Point: "Handles suboptimal actions..." is vague. Table 1 misses FAWAC/QIPO.
> > > >
> > > > We understand the reviewer's concern about the vagueness in our original characterization and the missing references. To improve clarity and precision, we have revised our comparison strategy in two ways:
> > > >
> > > > 1) Refined Table 1: We have narrowed the scope of Table 1 to focus exclusively on the Behavior-Regularized Actor-Critic (BRAC) family. This allows for a more meaningful comparison of specific algorithmic traits. Accordingly, we have removed IQL from this table and renamed it "Overview of regularization mechanisms within the BRAC framework."
> > > >
> > > > 2) New Figure 5 in the related work section: To address the broader landscape of prior work (including FAWAC and QIPO), we have introduced a new figure that visually highlights the architectural differences of our method relative to other approaches. This allows us to acknowledge these important baselines without cluttering Table 1.

---

> > > > > ### Author Response · Authors · 2025-11-27
> > > > > **Response to reviewer AGZW Part 2**
> > > > >
> > > > > ## Remark 3, Part 2. Unclear: Bidirectional Guidance
> > > > > >  Reviewer Point: confused on what bidirectional guidance means here ...difficulty parsing the description... it seems that QIPO also exhibits bidirectional guidance.
> > > > >
> > > > > We warmly thank the reviewer for sharing these concerns, which have helped us clarify the method and improve the paper. To resolve this confusion, in the second revision, we added a schematic overview Fig. 5 that visually distinguishes the components and respective interactions in GFP compared to prior work.
> > > > >
> > > > > In particular, Fig. 5f illustrates the concept of bidirectional guidance. The core contribution of GFP lies in its structure with **two policies that depend on each other**: it combines weighted BC and behavior regularized policy gradients. While the exact weighting function used within the weighted BC component has an impact, we view this as a design choice within the broader framework. The ablation study with $g_\eta^{AWR}$, section B.2, changes only the weighting function while keeping the general structure as illustrated Fig. 5f.
> > > > >
> > > > > Additionally, we have reworked the paragraph "Positioning GFP within prior work" and the sentence quoted by the reviewer. Our intention was to clarify that FAWAC and QIPO employ weighted BC as their _primary_ policy extraction method, and consequently, the weighted BC policy serves as their _final_ actor (see Fig. 5d). As the reviewer correctly notes, "both QIPO and VaBC rely on actor samples to compute the weights", however, in our case, the VaBC policy is not the final actor. GFP _primary_ actor is the one-step policy trained via behavior regularized policy gradients, which uses VaBC as a regularizer. This structural combination of two distinct policy extraction methods distinguishes GFP from previous approaches.
> > > > >
> > > > >
> > > > > ## Remark 2, part 1. AWR Formulation and Stochasticity
> > > > > > Reviewer Point: In Eq 11, the baseline is $Q_{\phi}(s, a_{\text{actor}})$. This is different from Peng et al... $a_{\text{actor}}$ is stochastic... optimum may not lead to correct policy.
> > > > >
> > > > > We appreciate the reviewer’s detailed examination of the baseline formulation. As clarified in our reply to remark 3 above, the specific weighting function inside the weighted BC component is a design choice. The original GFP weighting function is inspired by a soft-max weight between $Q(s,a)$ and $Q(s,a_\pi)$ (see Sec 3, step 3).
> > > > >
> > > > > We chose to rely on the stochastic estimator, accepting its variance, to avoid the approximation error and complexity associated with training a separate $V(s)$ network. This strategy is supported by modern architectures like SAC v2 [1], which noted: "In (Haarnoja et al., 2018c) we introduced an additional function approximator for the value function, but later found it to be unnecessary."
> > > > >
> > > > > In our controlled ablation study with $g^{AWR}$, we chose to keep the same architecture to ensure we were comparing the _weightening_ functions in isolation. However, we agree with the reviewer that investigating a separate V-network could be a valuable future direction, and we have added a sentence to Sec B.2 to highlight this point.
> > > > >
> > > > >
> > > > >
> > > > > ## Remark 2, part 2. Tuning the Bellman target for GFP-AWR
> > > > >
> > > > > > Specific Question: for your results in Table 7, did you tune GFP-AWR to use either the standard target or $y^{VaBC}$?
> > > > >
> > > > > Yes, we confirm that for the GFP-AWR ablation, we evaluated both the standard target and the conservative $y^{VaBC}$ target. The results reported in Table 7 reflect the best-performing configuration for each environment to ensure a fair comparison.
> > > > >
> > > > > We acknowledge that this level of detail was missing in the first revision. We have now added a sentence in Sec B.2 clarifying what target has been found for each environment with GFP-AWR.
> > > > > Specifically, we found that GFP-AWR performed best using $y^{VaBC}$ on the cube, humanoidmaze, and puzzle environments, similarly to GFP except on the puzzle environments, where for GFP both targets lead to similar results.
> > > > >
> > > > >
> > > > > [1] Haarnoja, T., Zhou, A., Hartikainen, K., Tucker, G., Ha, S., Tan, J., Kumar, V., Zhu, H., Gupta, A., Abbeel, P., & Levine, S. (2018). Soft Actor-Critic Algorithms and Applications. ArXiv, abs/1812.05905.

---

### Official Review · Reviewer_Zzn2 · 2025-10-28

**Soundness:** 3
**Presentation:** 3
**Contribution:** 2
**Rating:** 6
**Confidence:** 4

**Summary:**

This paper introduces a framework for performing offline RL using flow policies. Compared to prior such as FQL, which first trains a BC policy as a flow network, then uses this policy to distill a one-step policy which is then trained via Q-learning, this work also connects the Q network back to the flow network. Specifically, an AWR-style weighting is used on the loss, weighting higher-advantage actions. In essence this creates *two* feedback mechanisms for the learned Q function -- it affects the flow network (through advantage-weighting) and also the resulting one-step policy (through the DDPG loss). Experiments show that this method reliably outperforms prior work on a wide range of experiments.

**Strengths:**

This paper presents a clean framework with empirically sound results. The idea of using the AWR weighting to train the flow network is a neat way to achieve policy improvement wrt a learned Q function. It is useful computationally that the TD errors for Q-learning can be calculated using the distilled one-step policy, avoiding the need for an expensive ODE integration during training.

In terms of clarity and quality, the paper reads well and is easy to follow. See weaknesses section for additional points here.

The significance of this paper is boosted by its strong empirical performance and wide evaluation. The paragraph describing how the results in this work were re-run for previous baselines, and in fact improved upon them, lends credibility to the evaluation. In addition, the scaling curves with relation to temperature in Figure 3 clearly show that the intuition for the temperature term is correct in practice.

**Weaknesses:**

As this framework in essence uses two methods of feedback from the Q function, the paper would greatly benefit from a clear analysis on the effects of both of these terms. For example, two clear baselines are the same method where either the flow policy is not weighted (or if this is exactly FQL, some text making this clear) and where the one-step policy does not utilize the DDPG loss.

Building on the above, an analysis as to why the two feedback methods are complimentary would take this paper one step forwards. As of the current submission, it is shown that empirically it is useful to use both, but it is unclear on a deeper level why this might be true. The paper hints that "there is no fundamental justification for preferring one divergence or distance metric over another", and more investigation into this would be potentially fruitful.

**Questions:**

In the experiments, which version of the TD learning objective is used? Equation 7 shows that actions can be taken from either the distilled policy or the flow policy. Presumably taking actions from the flow policy is computationally expensive as an ODE integration is required.

In equation 6, the notation to use pi_c is somewhat confusing.

Does the optimal temperature change if the BC weighting (lambda) is adjusted, and vice-versa? What does the landscape of these hyperparameters look like, and how sensitive is the method to both of them?

It is interesting that both the one-step policy and the VaBC policy can be used for evaluation, as mentioned in table 2. What do the average scores for the VaBC policy look like?

---

> ### Author Response · Authors · 2025-11-24
> **Response to reviewer Zzn2's initial review : part 1**
>
> We thank the reviewer for their insightful comments and for recognizing our framework's cleanliness and empirical strength. The authors appreciate that the reviewer highlighted the two feedback mechanisms (AWR-style weighting on bc flow + DDPG on the one-step policy) and that our effort to re-run and improve upon baselines lent credibility to our evaluation.
>
> In the revised manuscript we reframed our method to better highlight that GFP uses two policy extraction methods, as the reviewer rightly pointed out, and we reworked Sec 5 to better position GFP within prior works. Modifications are listed in the general answer to all reviewers. We are grateful to the reviewer, as we believe the reviewer's questions regarding the two feedback methods helped us improve the paper.
>
>
> ### 1. Response to Weaknesses: Analyzing the two feedback mechanisms
>
> > **Reviewer Point:** The paper would benefit from analyzing the effects of the two feedback terms (Weighting vs. DDPG loss)
>
> 1. The Weighted BC Flow:
>
> The AWR-style weighting ensures that the multi-step flow policy focuses its probability mass on high-value regions present in the dataset.
>
> **Isolation of effect:** If we remove this weighting (i.e., use unweighted BC Flow + DDPG), our method effectively reverts to **FQL**, as the reviewer said. We have included a sentence clarifying this connection in the paper (end of Sec 5). As shown in our results (Table 2 and Figure 3), GFP consistently outperforms FQL. In particular, we added a few tasks for the rebuttal, and for instance, on the cube-triple-noisy environment GFP has an average score of 24.5 compared to 3.5 for FQL. This gap confirms the contribution of the weighted flow mechanism.
>
> **Note**: GFP weighting was inspired by a soft-max approach, but the reviewer rightly pointed out a connection with AWR-style weighting. We added a paragraph at the end of Sec 3 to clarify this relation with AWR, and we conducted new experiments on 65 tasks to compare GFP soft-max style weighting to AWR-style (in the appendix, Sec B.2).
>
> 2. The DDPG Loss:
>
> The DDPG loss ensures that the distilled one-step policy $\pi_\theta$ maximizes the Q-function locally while staying close to the high-value mode identified by the multi-step bc flow policy.
>
> **Isolation of effect:** If we rely solely on the weighted bc flow without the DDPG loss, the method reduces to a weighted BC approach. Without the DDPG term, the flow policy would become the primary policy, with the one-step policy only distilling for faster inference. This would imply using the flow policy to compute the targets in the critic update and to compute the weighting term. It results in an equivalent to **FAWAC** (Flow Weighted Advantage Actor-Critic) [1], with a purely distilled actor, recovering at best FAWAC performance. In the revised paper, we now include a comparison to 10 prior offline RL works, including FAWAC, on 50 tasks previously evaluated by [1] (see Fig. 3a and Tab 10).
>
> **Conclusion**: By comparing our method against FQL (Unweighted BC Flow + Distillation) and now also FAWAC (Weighted Flow + No Distillation with DDPG loss), we provide an ablation analysis of both mechanisms. On average over the 50 tasks evaluated by [1], GFP achieves a score of 51.8, compared to 43.5 for FQL and 16.1 for FAWAC.

---

> ### Author Response · Authors · 2025-11-24
> **Response : part 2**
>
> ### 2. Response to Specific Questions
>
> > Q1: In the experiments, which version of the TD learning objective is used? (Distilled vs Flow)
>
> For the TD learning objective (Eq. 7), by default, we use the standard Bellman objective in our experiments. We only use the modified objective for the cube and humanoidmaze-medium environments. We thank the reviewer for asking for clarification, as it was previously only implicitly mentioned. We now specify it in the hyperparameter table, Tab 4, and in the appendix Sec B.1.
>
>
> > Q2 : In equation 6, the notation to use pi_c is somewhat confusing.
>
> We had introduced the notation $\pi_c$ as a shorthand to refer generically to both the flow policy ($\pi_\omega$) and the distilled policy ($\pi_\theta$). We thank the reviewer for pointing out the risk of confusing readers. We have revised the manuscript to remove $\pi_c$ and instead explicitly refer to $\pi_\omega$ and $\pi_\theta$ individually to ensure clarity.
>
> > Q3: What does the landscape of these hyperparameters look like? Does optimal temperature change if BC weighting is adjusted?
>
> Following the reviewer's suggestion, we conducted a new sensitivity analysis on both the temperature $\eta$ and the BC weight $\alpha$. We added a new subsection in the appendix, Sec B.5 and Fig 7, with a reference to it Sec 4.2. For 12 tasks, we took the task-specific hyperparameters $(\alpha,\eta)$ and we tried all 8 variations by increasing or decreasing either or both alpha and eta.
>
> From this study, we made two remarks, first, GFP is primarily sensitive to $\alpha$ and less sensitive to $\eta$, in line with the ReBRAC conclusion that $\alpha$ is the most important parameter of BRAC methods. As a comparison, we reevaluated FQL sensitivity to $\alpha$. This also justifies our hyperparameter search methodology, we first fix $\eta = 10^{-3}$ and sweep over $\alpha$, and we sweep over $\eta$ only once $\alpha$ is chosen.
>
> Second, we observe a small correlation between the two: when $\alpha$ increases (i.e. stronger regularization), it is better to decrease $\eta$ (stronger filtering). Validating the intuition that when the actor is forced to stay close to the flow policy, the latter should strongly focus on cloning the best actions from the dataset.
>
>
> > Q4: interesting that both the one-step policy and the VaBC policy can be used for evaluation. What do the average scores for the VaBC policy look like?
>
> We have evaluated the VaBC flow policy as a byproduct of our method, and reported the individual scores. We originally did not include VaBC average performance to highlight that it is not GFP primary policy. We have now added the VaBC average scores Tab 2.

---

### Official Review · Reviewer_Aadb · 2025-10-30

**Soundness:** 3
**Presentation:** 2
**Contribution:** 3
**Rating:** 6
**Confidence:** 4

**Summary:**

The paper proposes Guided Flow Policy (GFP), an offline RL method that couples (i) a multi-step flow-matching policy trained by value-aware behavior cloning (VaBC) and (ii) a distilled one-step actor guided by a critic. The key idea is a bidirectional guidance: the actor and critic guide the flow model to preferentially clone high-value dataset actions (rather than all actions), while the flow model regularizes the actor to remain within the support of high-value dataset transitions. A temperature-controlled guidance term g_η modulates how selective the cloning is. Extensive experiments across 129 tasks spanning OGBench, Minari, and D4RL show that GFP consistently outperforms pervious methods.

**Strengths:**

The method is elegant and easy to follow. It cleanly integrates flow matching into the BRAC framework: a value-aware flow policy shapes the action distribution, and a distilled one-step actor provides fast inference. The bidirectional guidance narrative is intuitive, and the overall training loop is straightforward to implement.
The empirical evaluation is thorough. The paper reports results on 129 tasks spanning OGBench, Minari, and D4RL, providing broad coverage and reducing the risk of cherry-picking. Across these diverse settings, GFP demonstrates consistently strong performance.
Finally, the paper offers a useful sensitivity analysis of the temperature η. The study clarifies how guidance sharpness trades off selectivity versus diversity, giving practitioners a clear knob to choose stable, high-performing settings.

**Weaknesses:**

First, the novelty is moderate. The method still fits squarely within the BRAC family: it combines an expressive flow-matching policy with value-aware cloning and then distills into a one-step actor. While the synthesis is thoughtful and practically useful, the conceptual step beyond prior behavior-regularization plus expressive policy models feels incremental.

Second, the paper’s structure could be clearer, especially in the Experiments section. Main results and analysis are interleaved, which makes it harder to extract the headline takeaways before diving into diagnostics. I recommend separating “Main Results” (per benchmark, with concise aggregates) from “Analyses & Ablations.” In particular, if my understanding is correct, Section 4.1 “Suboptimal datasets” functions primarily as a temperature sensitivity study on η; the current title is confusing. Retitling to something like “Sensitivity to Temperature on Noisy/Suboptimal Data” would improve readability.

Third, the role of distillation isn’t entirely clear in the current narrative. Table 3 reports VaBC-only results, but the training pipeline still relies on a distillation phase, which makes it hard to tease apart how much of the gain comes from the value-aware flow versus the student actor. A clean ablation that removes distillation end-to-end (i.e., deploy the flow directly), paired with a short intuition for when distillation helps would clarify its necessity and practical benefit.

**Questions:**

See weaknesses.

---

> ### Author Response · Authors · 2025-11-24
> **Response to reviewer Aadb's initial review**
>
> We thank the reviewer for the positive assessment of our work and for recognizing the intuitive nature and empirical strength of our method.
>
> ### 1. Response to Weaknesses 1: Novelty is moderate
> > **Reviewer Point:** method still fits squarely within the BRAC family .. conceptual step [..] feels incremental
>
> We agree with the reviewer that GFP belongs to the BRAC family, and is made of existing components. We apologize for not having clarified our contributions in the original submission enough, and acknowledge a lack of positioning with respect to prior works. During the rebuttal, we modified our paper to clarify GFP core contributions. Modifications are listed in the general answer to all reviewers.
>
> To respond to the reviewer's concern of moderate novelty: GFP indeed belongs to the BRAC family, but the key contribution is to add value-awareness in the regularization component of the actor. Effectively combining two of the predominant policy extraction methods in offline-RL: weighted BC (for the flow policy) and behavior regularized policy gradients.
>
> These two components are interdependent through our bidirectional guidance, and we reworked the paper to clarify this combination as our main contribution. We believe that our extensive benchmarks assess the effectiveness of our approach. In particular, through this guidance, we train a value-aware flow policy without backpropagating through time (whereas for instance, FQL trains a reward-agnostic flow matching and then uses it to train a one-step policy).
>
> To address this weakness raised by the reviewer:
> - We restructured the paper and especially Sec 5 on related works, to clarify GFP position within prior work.
> - We added a comparison to 10 offline-RL methods on 50 tasks, showing how GFP stands out compared to all considered prior works (performance profiles Fig 3a, and full results Tab 10)
> - We extended our study to 144 tasks to reaffirm GFP scaling abilities to very challenging tasks, e.g. on the cube-triple-noisy environment GFP achieves an average score of 24.5 compared to 4.8 for IQL, 5.2 for ReBRAC and 3.5 for FQL (Tab 2 and Tab 9).
>
>
>
> ### 2. Response to Weaknesses 2: Paper structure
> > **Reviewer Point:** results and analysis are interleaved .. recommend separating “Main Results” from “Analyses & Ablations.”
>
> We thank the reviewer for this valuable suggestion, in the revised version of the paper we reworked the experiment section in three 3 subsections: (i) Main results (first on 50 tasks against 10 prior works and then on 144 tasks); (ii) Analysis of the temperature (where we additionally did a sensitivity analysis on both alpha and eta); (iii) Re-evaluation of prior works. We believe the reviewer's comment helped us improve the flow and clarity of the section.
>
> ### 3. Response to Weaknesses 3: Role of distillation unclear
> > **Reviewer Point:** role of distillation isn’t entirely clear... A clean ablation that removes distillation end-to-end... would clarify its necessity.
>
> This is a crucial point, and we appreciate the opportunity to clarify the importance of the distillation phase. GFP combines two policy extraction methods, (i) behavior regularized policy gradient for the actor (ii) weighted behavior cloning for the flow policy.
> If we were to remove the one-step actor, it would mean losing the behavior regularized policy gradient component, and imply using the flow component to compute the targets in the critic update and using a value estimate to compute the baseline for the guidance, relying solely on weighted BC. The resulting algorithm becomes structurally equivalent to FAWAC (Flow Weighted Advantage Actor-Critic) [1], with the minor difference that FAWAC explicitly trains a state-value function for the baseline.
>
> To address this question raised by the reviewer in the revised version, we include a comparison to 10 prior works on the 50 tasks evaluated by [1], including FAWAC, which is equivalent to the ablation study recommended by the reviewer. Tab 10 reports the corresponding results, FAWAC has an average score of 16.1 over the 50 tasks, while GFP achieves a score of 51.8. Fig 3a shows that FAWAC is one of the least effective approaches. This can be explained following [2] conclusion that behavior regularized policy gradient is a more effective policy extraction method than weighted BC. GFP is the only method to recover the advantage of both policy extraction methods with flow matching.
>
> [1] Park, S., Li, Q., & Levine, S. (2025). Flow q-learning. arXiv preprint arXiv:2502.02538.
>
> [2] Park, S., Frans, K., Levine, S., & Kumar, A. (2024). Is value learning really the main bottleneck in offline RL?. Advances in Neural Information Processing Systems, 37, 79029-79056.

---

### Author Response · Authors · 2025-11-24
**Official response to all initial reviews**

The authors thank all the reviewers for their constructive feedback and for recognizing the soundness of our method and the rigor of our large-scale evaluation. We modified the submission to address the reviewers' concerns. Following their suggestions, we conducted new experiments listed below. We thank the reviewers, as we firmly believe these modifications and additional experiments strengthen the paper.

Overall, the main concern raised was about the positioning of GFP within prior works, which obscured the paper's novelty. We appreciate this insight, as addressing it has allowed us to refine the framing of GFP and more clearly articulate its distinct contribution compared to the existing literature.

We conducted and integrated the following new experiments:
- We added a comparison of GFP to 10 offline-RL baselines, on the 50 tasks previously evaluated by FQL's authors, showing how GFP stands out compared to all considered prior works (performance profiles Fig 3a, and full results Tab 10).
- We extended our full study to 144 tasks, including 5 pixel-based tasks. In particular, on the challenging cube-triple-noisy GFP achieves an average score of 24.5 compared to 4.8 for IQL, 5.2 for ReBRAC, and 3.5 for FQL (Tab 2 and Tab 9).
- To study how sensitive GFP is to $\alpha$, the behavior cloning coefficient, and $\eta$, the guidance temperature, we systematically tested variations around the task-specific chosen $(\alpha,\eta)$ parameters. The analysis was done on 12 tasks, with comparisons to FQL sensitivity to $\alpha$, see the appendix Sec B.5 and Fig. 7.
- Following the suggestion to try directly using an advantage weighting term (similar to AWR, but using the actor to compute the baseline), we benchmarked GFP soft-max weight against an AWR-style weight on 65 tasks. See the new explanations of how GFP relates to AWR at the end of Sec 3 and the ablation study in the appendix, Sec B.2 and Tab 7.

Along with these experiments, we made the following modification to the paper:
- Throughout the paper, we made modifications to clarify GFP core contribution as a framework combining two policy extraction methods predominant in offline RL: weighted BC and behavior regularized policy gradients. Integrating value-awareness in the regularization term of a behavior regularized actor critic method (in particular impacting the end of the introduction and the conclusion). We thank the reviewers for seeing the conceptual simplicity of GFP as a strength.
- Section 1: We modified Tab 1 to clarify the "handle suboptimality" heading.
- Section 3 step 3: We clarified how GFP's weighted BC component relates to AWR.
- Section 4: Following reviewer Aadb rightful suggestion, we restructured the experimental section, in three 3 subsections: (i) Main results (first on 50 tasks against 10 prior works and then on 144 tasks against the 3 selected ones); (ii) Analysis of the temperature (where we additionally mention the new sensitivity analysis on $\alpha$ and $\eta$); (iii) Re-evaluation of prior works.
- Section 5: We completely reworked the related work section to better position GFP within prior works, especially to clarify GFP relation to pure weighted BC methods (e.g. FAWAC). We thank the reviewers for having pointed out this section as the main issue in the original submission.
- We integrated all the new experiments in the appendix, and restructured the order in which tables are presented to help readability.

We have responded to each reviewer individually below with specific answers and technical clarifications. We believe we have addressed all their concerns and recommendations, so we look forward to the discussion.

---

### Author Response · Authors · 2025-11-27
**Second revision, new figure for the related work section, Fig 5, page 10**

Following reviewer AGZW's response earlier this week, we made a second revision to our paper. In addition to the new experiments, new figures and all changes listed in the official comment below, **in the second revision, a new figure has been added to the related work section** to illustrate an overview of offline-RL methods, see Fig 5, page 10. The related work section has been modified accordingly, and Table 1 has been reworked.

---

### Author Response · Authors · 2025-12-03
**Summary of rebuttal phase**

To the Area Chair,

We sincerely appreciate the additional time and effort the AC dedicates to reviewing ICLR submissions this year, due to the unfortunate leaks that perturbed the rebuttal phase. We would like to provide a brief overview of the rebuttal phase, particularly regarding the constructive exchange with reviewer AGZW.

## Initial reviews
The initial scores the paper received were 6, 6, 6, and 2. All reviewers highlighted the method's soundness and our rigorous evaluation. Notably, even reviewer AGZW, who initially assigned a score of 2,  was aligned with the accepting reviewers in their comments.
From the beginning, reviewer AGZW noted that our comparisons were "very convincing" and that the method was "concise and clear" (grading soundness=3, presentation=3, contribution=2). The main issue identified by the reviewers was on the positioning with respect to prior work, which led to questions about the novelty of our work, with reviewer AGZW considering this issue to be critical.

## First revision
During the rebuttal phase, we conducted additional experiments (comparing against more methods, studying sensitivity, testing more challenging tasks, and performing an ablation study on the guiding function), we revised the manuscript comprehensively, and in particular, we completely reworked the related work section. Following our first responses, reviewer Zk33 confirmed they "don't have any outstanding concerns", while reviewers Aadb and Zzn2 unfortunately did not have time to respond.

## Response from reviewer AGWZ and second revision
Following our first revision adding extensive comparisons to prior work, reviewer AGZW acknowledged the technical novelty and raised their score to 4. The reviewer requested clarification on three additional points to fully justify the method's design. We are grateful for their continued engagement, and we made a second revision to definitively clarify these points. In particular, we made a new figure for the related work section. Unfortunately, due to the rebuttal phase freeze, reviewer AGZW was not able to react anymore.

## Conclusion
We believe our discussion log shows a clear trajectory toward convergence. We are sincerely grateful to the reviewers, whose comments have helped us to improve the paper significantly, especially, but not only, regarding our positioning. The main modifications are listed in the two general comments below.
We again thank the Area Chair for the time and consideration in assessing our paper.

Note: we submitted a third revision only to fix some typos.

---

### Meta-Review · Area_Chair_bFWh · 2026-01-06

**Summary:**

This paper proposes Guided Flow Policy (GFP), an offline RL method that combines a value-aware multi-step flow policy (trained with value-aware behavior cloning / VaBC) with a distilled one-step actor trained with a critic, forming a bidirectional guidance loop that focuses cloning on high-value dataset actions while regularizing the actor to stay within high-value support. Reviewers found the approach elegant and the empirical coverage unusually broad (OGBench/Minari/D4RL, 100+ tasks), but key concerns were moderate novelty (seen as an AWR-style weighting + BRAC/FQL-style distillation), unclear role/necessity of distillation, clarity/structure of experiments, fairness and tuning details (e.g., conservative targets only available to GFP), and stronger positioning vs related weighted-BC diffusion/flow methods (FAWAC/QIPO) and broader baselines.

**Reviewer Concerns:**

The rebuttal addressed most major issues: the authors restructured experiments into “main results vs analyses,” expanded related work and clarified that VaBC is not claimed as first-ever advantage weighting but as a coupled regularizer inside BRAC with bidirectional interactions, and added broader baseline comparisons (10 methods on 50 tasks) plus expanded evaluations (up to 144 tasks including challenging/noisy and pixel-based tasks). They also clarified distillation’s role via ablations/links to FQL (unweighted flow + distillation) and FAWAC-like (weighted flow without behavior-regularized policy gradient), added sensitivity studies over temperature and BC weight, and responded to fairness concerns by tuning critic target aggregation choices across methods and environments. Remaining concerns are relatively minor: one reviewer still questioned deeper justification for why VaBC is the right design and some details of AWR-style formulation, but those were partially mitigated with added ablations, clarified tuning of conservative targets, and revised positioning/figures.

**Reviewer Scores:**

Aadb likely stays 6→6 (concerns about structure/distillation were addressed; novelty still “moderate” but evidence strengthened). Zzn2 likely 6→7 (requested ablations and clearer analysis of the two feedback mechanisms were added). AGZW increased from 2→4 explicitly (improved positioning + added comparisons), and may remain around 4 after the follow-up clarifications. Zk33 likely stays 6→6 (explicitly maintained slightly positive score). Overall sentiment shifts to mildly positive with strengthened evidence.

---

### Decision · Program_Chairs · 2026-01-26

Accept (Poster)